# Ancient DNA and deep population structure in sub-Saharan African foragers

Mark Lipson[1,2,40 ✉], Elizabeth A. Sawchuk[3,4,40 ✉], Jessica C. Thompson[5,6 ✉], Jonas Oppenheimer[7], Christian A. Tryon[8,9,10], Kathryn L. Ranhorn[6], Kathryn M. de Luna[11], Kendra A. Sirak[1,2], Iñigo Olalde[1,12], Stanley H. Ambrose[13], John W. Arthur[14], Kathryn J. W. Arthur[14], George Ayodo[15], Alex Bertacchi[5], Jessica I. Cerezo-Román[16], Brendan J. Culleton[17], Matthew C. Curtis[18], Jacob Davis[19], Agness O. Gidna[20], Annalys Hanson[21], Potiphar Kaliba[22], Maggie Katongo[23,24], Amandus Kwekason[20], Myra F. Laird[25], Jason Lewis[4], Audax Z. P. Mabulla[26], Fredrick Mapemba[22], Alan Morris[27], George Mudenda[24], Raphael Mwafulirwa[28], Daudi Mwangomba[29], Emmanuel Ndiema[30], Christine Ogola[30], Flora Schilt[31], Pamela R. Willoughby[3], David K. Wright[32,33], Andrew Zipkin[34], Ron Pinhasi[35,36], Douglas J. Kennett[37], Fredrick Kyalo Manthi[30], Nadin Rohland[1], Nick Patterson[2], David Reich[1,2,38,39 ✉] & Mary E. Prendergast[1,23 ✉]

Multiple lines of genetic and archaeological evidence suggest that there were major demographic changes in the terminal Late Pleistocene epoch and early Holocene epoch of sub-Saharan Africa[1-4]. Inferences about this period are challenging to make because demographic shifts in the past 5,000 years have obscured the structures of more ancient populations[3,5]. Here we present genome-wide ancient DNA data for six individuals from eastern and south-central Africa spanning the past approximately 18,000 years (doubling the time depth of sub-Saharan African ancient DNA), increase the data quality for 15 previously published ancient individuals and analyse these alongside data from 13 other published ancient individuals. The ancestry of the individuals in our study area can be modelled as a geographically structured mixture of three highly divergent source populations, probably reflecting Pleistocene interactions around 80–20 thousand years ago, including deeply diverged eastern and southern African lineages, plus a previously unappreciated ubiquitous distribution of ancestry that occurs in highest proportion today in central African rainforest hunter-gatherers. Once established, this structure remained highly stable, with limited long-range gene flow. These results provide a new line of genetic evidence in support of hypotheses that have emerged from archaeological analyses but remain contested, suggesting increasing regionalization at the end of the Pleistocene epoch.

Models for the expression of human behavioural complexity during the Late Pleistocene (around 125–12 thousand years ago (ka)) often invoke demographic change[1,2]. By around 50 ka, technological innovations and symbolic behaviours (such as ornaments, bone tools, pigments and microliths) that were present earlier in the Middle Stone Age (MSA) become more consistently expressed across sub-Saharan Africa[4,6,7]. Archaeologists refer to this as the transition to the Later Stone Age (LSA)[1,7–9]. By around 20 ka, these material culture components were nearly ubiquitous, but regionally diverse. One explanation is that people began living in larger and/or more connected groups, with variations in population size and connectivity driving differences in material culture across space and time. Given the morphological variation among Late Pleistocene skeletons, interactions may have involved deeply structured populations[2,10], consistent with some population history models based on genetics[3].

The advent of genome-wide ancient DNA (aDNA) technology holds promise for better understanding major changes in material culture and hypothesized demographic shifts among ancient African foragers (Supplementary Notes 1, 2). Compared to elsewhere, especially Europe, there has been little genomic investigation of ancient African peoples. Previously available aDNA sequences from sub-Saharan African foraging contexts[11–14], despite being relatively recent (younger than about 9 ka), provide evidence of ancient genetic structure that has since been disrupted by demographic transformations (such as the spread of food production, as well as colonialism, imperialism, enslavement and modern sociopolitical reorganization). The structure of ancient populations cannot be robustly reconstructed based solely on genetic data from present-day people.

Here we present new genome-wide aDNA data and radiocarbon dates from three Late Pleistocene and three early to middle Holocene individuals associated with LSA technologies at five sites in eastern and

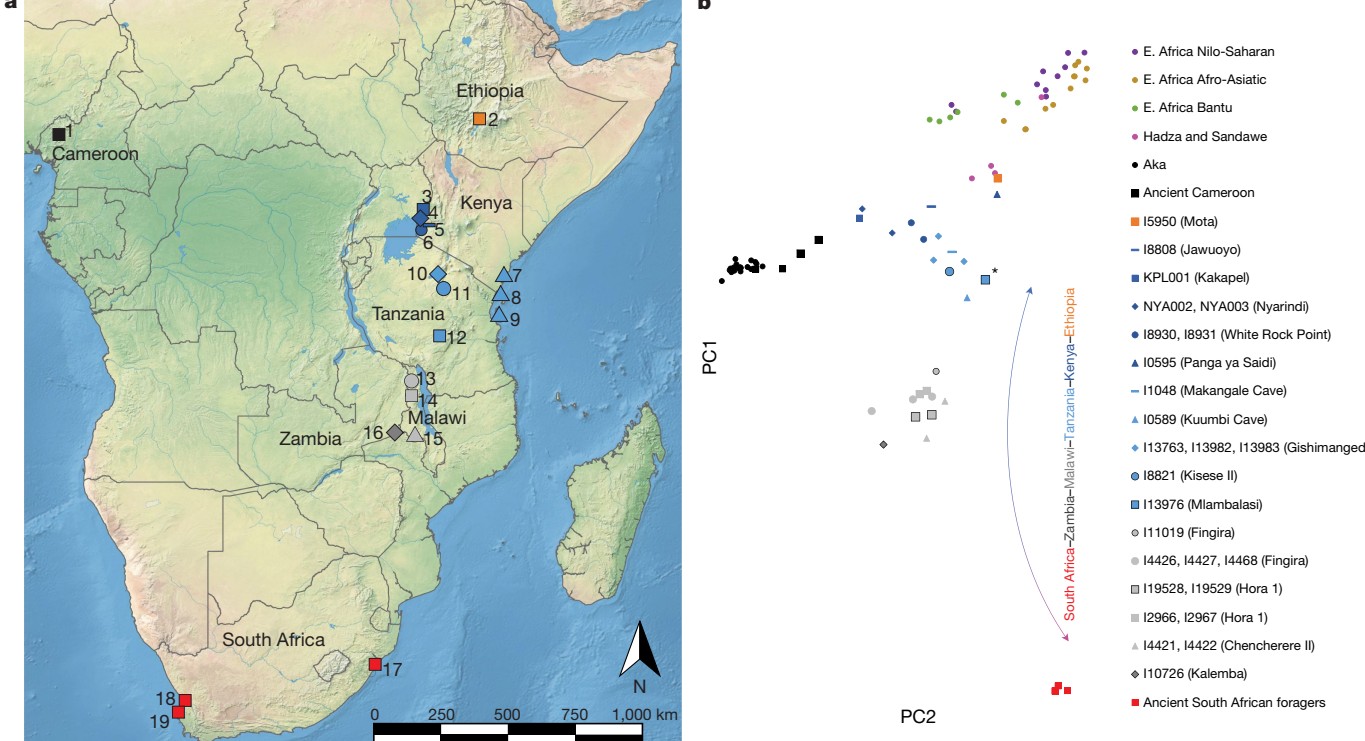

**Fig. 1 | Locations of the individuals analysed and PCA analysis. a**, Locations of individuals analysed in this study. The shapes and colours of the symbols correspond to the PCA in **b**. 1, Shum Laka; 2, Mota Cave; 3, Kakapel RS (Rockshelter); 4, Nyarindi RS; 5, Jawuoyo RS; 6, White Rock Point; 7, Panga ya Saidi; 8, Makangale Cave; 9, Kuumbi Cave; 10, Gishimangeda Cave; 11, Kisese II RS; 12, Mlambalasi RS; 13, Fingira; 14, Hora 1; 15, Chencherere II; 16, Kalemba RS; 17, Ballito Bay; 18, Faraoskop RS; 19, St Helena. **b**, PCA results. Axes were computed using present-day groups from eastern (Dinka pastoralists), southern (Ju|'hoansi foragers) and central Africa (Mbuti foragers). Small circles represent present-day individuals; other symbols represent ancient individuals (larger points corresponding to earlier individuals and black outlines to newly reported individuals). The lowest-coverage individual (from Mlambalasi), shown with an asterisk, has the most uncertain position. The base map in **a** is from Natural Earth (https://www.naturalearthdata.com). E., east.

south-central Africa: Kisese II and Mlambalasi Rockshelters in Tanzania, Fingira and Hora 1 Rockshelters in Malawi, and Kalemba Rockshelter in Zambia (Fig. 1a and Extended Data Table 1). Direct and indirect dates range from around 18 ka to 5 ka, doubling the time depth of aDNA reported from sub-Saharan Africa. We analyse these data together with the published sequences of 28 other ancient African individuals recovered from contexts spanning the past 8,000 years and largely associated with foraging at 17 sites in eastern, central and southern Africa. We also provide higher-coverage data for 15 of these individuals. Analysis of the ancient data together with sequences from present-day groups, aided by new statistical methods, enables a reconstruction of changes in regional- and continental-scale population structures among people who lived before the sweeping demographic changes of the past approximately 5,000 years. It also enables comparisons of Pleistocene forager population dynamics between the tropics and more temperate regions.

## The dataset

Of 31 samples (Supplementary Table 1), five petrous bones and one distal phalanx yielded aDNA sequences, which, after preparation of up to six libraries from each sample and enrichment for a panel of around 1.2 million single-nucleotide polymorphisms (SNPs), ranged in coverage from 0.001–3.2× (median, 0.06×) of targeted genome-wide SNP positions (Extended Data Table 1 and Supplementary Table 2). Additional archaeological and bioarchaeological information for these individuals is summarized in Supplementary Note 3. Direct $^{14}$C dates were attempted for the five petrous bones, but only two preserved sufficient collagen: Kalemba (I10726; 5,280–4,880 calibrated years before present (cal. BP), PSUAMS-4764) and Kisese II (I18821; 7,240–6,985 cal.

BP, PSUAMS-4718) (Supplementary Table 3 and Supplementary Note 4). Moreover, a new date was generated on enamel carbonate for a published individual from Hora 1 (I2966; previously estimated around 8,100 BP, now directly dated to 9,090–8,770 cal. BP, PSUAMS-5145). Individuals from Mlambalasi (I13976; about 20–17 ka) and Hora 1 (I19528, I19529; 17–14 ka) are well constrained to the Late Pleistocene based on multiple indirect dates (Supplementary Table 4 and Supplementary Note 3). One individual from Fingira (I11019) is represented by a distal phalanx that was recovered in isolation near the surface during excavation. This sample was too small to be both dated and assessed for aDNA; its age is constrained to around 6,200–2,300 cal. BP by association with direct dates on other human remains from the site. The 15 previously published individuals[11,13,15,16] (Supplementary Note 3) for which we increase sequence coverage include approximately 26× shotgun coverage for the individual from Mota Cave in Ethiopia[15] (I5950), enabling reliable calling of diploid genotypes (Extended Data Table 1, Methods and Supplementary Table 2). The authenticity of the new aDNA data was assessed through a combination of several criteria; detectable contamination was observed for only two samples (Methods, Extended Data Fig. 1a, Supplementary Table 2 and Supplementary Note 5). In Supplementary Table 5 and Supplementary Note 5, we report genotypes at SNPs associated with lactase persistence, sickle cell trait and the Duffy antigen, with derived alleles observed only at the DARC (Duffy) locus (four published individuals from Cameroon).

## Uniparental markers

All four newly reported males are similar to most published ancient foragers from this region of Africa in carrying the widely distributed

Y chromosome haplogroup B2 (Extended Data Table 1). Among the 23 individuals in our dataset with known mtDNA haplogroups, up to 14—almost all from Kenya and Tanzania—have haplogroups that are today associated with eastern Africa (Extended Data Table 1 and Supplementary Table 6). Eight individuals—all from Malawi and Zambia—have haplogroups that are associated with some ancient and present-day southern African people, specifically groups for whom foraging is the main mode of subsistence[17–20]. Two individuals from Malawi (I19529 from Hora 1, dating to about 16 ka and carrying L5b, and I4426 from Fingira, dating to about 2.3 ka and carrying L0f/L0f3) have eastern-Africa-associated haplogroups, whereas a different individual from Malawi (I2967 from Hora 1, dating to about 8.2 ka with L0a2/L0a2b) and possibly one from Kenya (I8930 from White Rock Point with L2a4) belong to lineages that are characteristic of present-day central African foragers (such as Mbuti and Aka). These results show that eastern and south-central Africa was home to, and an area of interaction among, diverse ancient foraging groups, and also that several of these haplogroup lineages were formerly more widespread than they are today.

## Three-way cline of genome-wide ancestry

For the bulk of our analyses, we used the genome-wide genotype data to gain insights into the ancestry of the ancient forager individuals and their connections to other groups. We performed a supervised principal component analysis (PCA) (Methods) in which we used three present-day groups—Ju|'hoansi (San) from southern Africa, Mbuti from central Africa and Dinka from northeastern Africa—to define a two-dimensional plane of variation, and projected all other individuals (ancient and present day) onto this plane (Fig. 1b). Consistent with previous studies[5,11,13,14], we observed an ancient latitudinal gradient of ancestry, represented at its northern extreme by an individual from around 4.5 ka from Mota Cave, Ethiopia, and its southern extreme by individuals from around 2 ka from South Africa. The newly reported individuals generally cluster with their geographical neighbours but extend documentation of the cline both geographically (southwest to Kalemba, at the corresponding extreme on PCA) and temporally (to a maximum of approximately 18–16 ka, with no apparent temporal subclusters). Furthermore, we found complexity in the cline in the form of deviations from a straight line: (1) the main direction of variation does not align with ancient southern African foragers; and (2) several individuals appear to shift in the direction of present-day and ancient central African foragers. Both observations may indicate that some of the ancient eastern and south-central African individuals sampled here trace part of their ancestry to groups that are related to foragers currently living in central Africa. Furthermore, (1) could indicate that the southern-African-related ancestry among the ancient individuals is only distantly related to present-day Ju|'hoansi and ancient southern African foragers.

We used allele-sharing tests (*f*-statistics) (Methods) to further investigate which individuals differed in their degree of relatedness to ancient South African foragers (AncSA) (Extended Data Table 1), the Mota individual or present-day Mbuti. Consistent with the PCA, most pairs of individuals from the same region (including from different time points) were nearly symmetric in their ancestry (|Z| < 3) (Supplementary Table 7). The exceptions were (1) excess affinity between Mbuti and KPL001 (Kakapel; max Z = 5.1); (2) excess affinity between AncSA and I0589 (Kuumbi Cave; max Z = 4.1); and (3) modest differences within Malawi and Zambia (max Z = 3.8). By contrast, well-powered cross-region statistics were highly significantly non-zero, for example, $f_4$(I8808 (Jawuoyo), I8821 (Kisese II); Mota, AncSA) > 0, Z = 7.8. We also used the qpWave program in ADMIXTOOLS to combine multiple *f*-statistic-based signals into a test for the number of distinct components of ancestry (relative to a specified outgroup set) present among the (sampled) ancient forager individuals (Methods). We found

that at least three sources are necessary ($P = 6.4 \times 10^{-14}$ for rejecting a two-source model) but, interestingly (with our available statistical power), that three sources are also sufficient (P = 0.73; four versus three sources P = 0.15), even with Mota, San (here, both Ju|'hoansi and ‡Khomani) and Mbuti among the outgroups. When we added the Mota individual to the test set, we found increased evidence for a fourth source, despite the less stringent outgroups (P = 0.07; four versus three sources P = 0.019) (Methods). This result could reflect a highly divergent ancestry component contributing to the Mota individual inferred in previous work[16]; additional lineages may also have been present among as-yet unsampled ancient individuals from these regions.

We attempted to estimate the dates of admixture (potentially involving any distinct sources of ancestry) for the ancient foragers using DATES[21]. With the caveat that our power is limited by data availability, we obtained only two robust estimates (Supplementary Table 8), both for previously published individuals, and both (given the additional results below) are probably connected to admixture from food producers: for I4421 (Chencherere II, no direct age, past approximately 5,000 years), a date of 10 ± 2 generations before the individual lived; and for I1048 (Makangale Cave, direct age, past approximately 1,500 years), 79 ± 24 generations before the individual lived.

## Inter- and intraregional relationships

Next, we modelled the ancestry of the ancient foragers in an admixture graph framework to test additional hypotheses concerning their relationships, aided by a new methodology to increase available information from low-coverage data (Figs. 2 and 3, Methods, Supplementary Notes 6 and 7 and Extended Data Figs. 2–5). In model 1, along with other populations, we included three geographically and genetically diverse ancient eastern and south-central African individuals with high sequencing coverage: I4426 (Fingira, about 2.5 ka), I8821 (Kisese II) and I8808 (Jawuoyo). On the basis of the results in the previous section, we hypothesized that they could be fit with mixtures of three ancestry components: one related to the Mota individual (representing an ancient group of foragers from the northern part of eastern Africa), one related to central African foragers (represented by present-day Mbuti) and one related to southern African foragers (represented by four ancient individuals from South Africa). Indeed, we obtained a good fit to the data in model 1 (max residual Z = 2.0), even when specifying identical sources for all three individuals, and the relative ancestry proportions were as expected: Mota-related ancestry decreased from north to south, and Jawuoyo (I8808) had the highest ratio of central-African-related ancestry to southern-African-related ancestry. Omitting any of the three components for any of the individuals results in a poor fit (Z ≥ 4.0) (Supplementary Note 6). As in ref.[16], we also estimated around 30% of a separate and deeply diverged 'ghost' ancestry component in the Mota individual (replicated here using new higher-coverage diploid whole-genome data).

When we added more individuals to create models 2 and 3 (max residuals Z = 3.0 and Z = 3.7), we found that the overall inferred structure and parameters were similar to those of model 1 (Supplementary Tables 9 and 10; see below for specific individuals and regions). The Mota-related and southern-African-related ancestry sources are inferred to split deeply along their respective lineages, meaning that, in some sense, they represent 'ghost' populations (without closely related sampled representatives). The central-African-related component is inferred to be closer to Mbuti (including an ancestral admixture event; Supplementary Note 6) than to Aka, and therefore to not split as deeply relative to the initial divergence of the central African forager lineage. Almost all of the additional significant allele-sharing signals that we observed beyond those in model 1 can be attributed to one of the three following causes (Supplementary Table 11): (1) excess relatedness at short-distance scales (see below); (2) admixture from pastoralists and/or farmers more recent than our period of focus (four individuals);

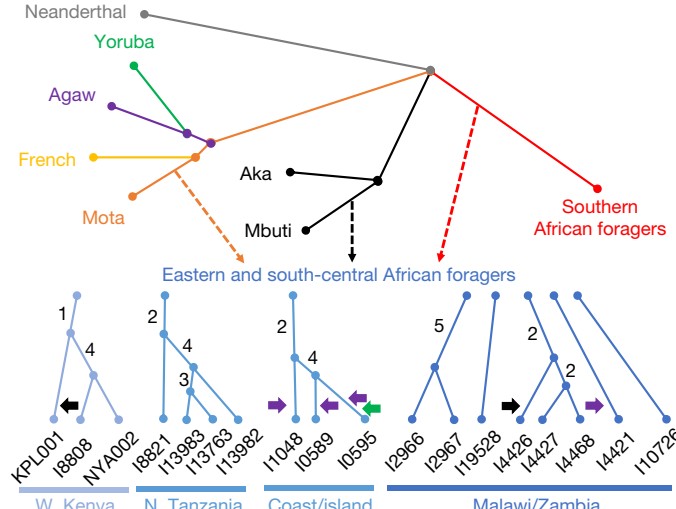

**Fig. 2 | Schematic of admixture graph results.** Branch lengths are not to scale. The arrows denote admixture events, with the three primary components of ancestry shown as dashed arrows, and other inferred gene flow as small solid arrows (with colours corresponding to related groups). Subclusters of ancient eastern and south-central African foragers reflect the inferred instances of excess relatedness among individuals, with internal branch lengths shown in genetic-drift units. Mixture proportions are shown in Fig. 3 and Supplementary Table 9 and the full results are shown in Extended Data Fig. 4. Individual laboratory numbers are shown at the bottom (Extended Data Table 1). N., north; W., west.

or (3) contamination (two individuals). In these cases, we adjusted our final model by (1) allowing shared history (that is, genetic drift) between the relevant individuals; (2) adding the inferred admixture events; or (3) incorporating extra admixture to represent the contamination source (Supplementary Note 6).

For sites in western Kenya, we found that all three individuals in model 3 have excess relatedness beyond the baseline expectation (Fig. 2). The individuals from Jawuoyo (I8808) and Nyarindi (NYA002/NYA003) are the closest, and they can be modelled with Mota-related, central-African-related and southern-African-related ancestry in respective proportions of about 62%, 19% and 19%, while the individual from Kakapel (KPL001) is inferred to have around 12% additional central-African-related ancestry (s.e. of approximately 2–4% with some assumptions) (Fig. 3 and Supplementary Note 6). For north-central Tanzanian sites, again all four individuals have signals of mutual excess allele sharing, with the three individuals from Gishimangeda (I13763, I13982 and I13983) being the closest. One of the three (I13763) shows excess relatedness to non-African individuals, which we interpret as evidence of a small proportion of contamination (Supplementary Notes 5 and 6); otherwise, all four can be fit as a clade with 54%, 12% and 34% Mota-related, central-African-related and southern-African-related ancestry, respectively. Similarly, the three island and coastal individuals (Makangale Cave I1048, Kuumbi Cave I10589, Panga ya Saidi I0595) display excess relatedness, with those from Kuumbi Cave and Panga ya Saidi closest to one another, and with 49% Mota-related, 12% central-African-related and 39% southern-African-related ancestry. These individuals also have ancestry admixed from populations that are associated with food production: Agaw-related for all three, plus western-African-related for Panga ya Saidi (I0595) (Supplementary Note 6).

In contrast to Kenya and Tanzania, we did not observe widespread signals of excess relatedness in Malawi and Zambia. After adjusting for ancestry proportions, most individuals within this geographical cluster are no more related to one another than they are to individuals

from Kenya and Tanzania. The only notable exceptions that we found among those in model 3 (Supplementary Note 6) were as follows: (1) among individuals from Fingira (I4426, I4427 and I4468), in particular, two dating to about 6.1 ka; and (2) between the individuals from 9–8 ka from Hora 1 (I2966 and I2967). However, other individuals separated by as little as 100–150 km (Fingira-Hora 1 and Chencherere II-Kalemba) can be fit well with independent mixtures of the same ancestry sources used across the entire study region, including some individuals around 700–1,500 km away. At the same time, the inferred ancestry proportions for the individuals from Malawi and Zambia are quite similar (about 20–30% Mota-related, 5–10% central-African-related and 60–70% southern-African-related), with significant (but small) differences observed for I4426 from Fingira (approximately 11% additional central-African-related ancestry), I4421 from Chencherere (approximately 4% ancestry related to pastoralists), I10726 from Kalemba (approximately 5% less Mota-related ancestry than in Malawi) and I2966 from Hora 1 (a small amount of contamination). We also built an alternative version of our model in which we specified the Malawi individuals as forming a clade descended from a shared three-way admixture event (plus small proportions of additional admixture for the aforementioned individuals) that had only a slightly worse fit—confirming the very similar ancestry proportions among the individuals—but that featured zero shared drift at the base of the clade and almost none on the internal branches (Supplementary Note 6 and Extended Data Fig. 6).

We examined the relationship between geographical distance and genetic relatedness using a new approach based on the residuals of a model assuming that there is no excess shared genetic drift—that is, we observed the similarity of genotypes within pairs of individuals relative to that predicted solely by differential proportions of the three ancestry sources (Methods). Using pairs of individuals from either Kenya and Tanzania, or Malawi and Zambia, together with inter-region pairs to plot the residuals as a function of distance, we found greater relatedness at short distances, but with different length scales for the decay of the fitted curves (about 60 km and about 3 km, respectively) (Extended Data Fig. 7a). Similar patterns are also observed if we omit pairs of individuals that were buried at the same site (Extended Data Fig. 7b). Thus, with the caveats that our sampling is not uniform and that not all of the individuals lived contemporaneously, we found on average that (1) individuals from the same or nearby sites are more closely related than predicted solely on the basis of the broad regional genetic structure, but (2) this relatedness extends only over short distances, particularly within Malawi and Zambia.

For a comparative perspective from contemporaneous ancient foragers in temperate environments, where there are more extensive available data, we performed similar analyses for individuals from Mesolithic Europe (*n* = 36, about 12–7 ka) (Methods, Supplementary Table 12 and Extended Data Fig. 7c, d). Both western and eastern/northern Europe also show a pattern of greater relatedness at shorter distances; western Europe is similar to Malawi and Zambia in that almost all of the signal comes from same-site pairs, but eastern/northern Europe has a substantially longer geographical decay scale.

Finally, we compared the ancient individuals to the present-day Sandawe and Hadza groups in Tanzania, who historically or recently practiced foraging lifeways. Previous studies have shown that the Hadza and Sandawe have distinctive ancestry from their neighbours, with unusually high proportions of ancestry related to ancient African foragers[11,13,14,22]. We built an extended version of model 2 including both groups (Extended Data Fig. 8 and Supplementary Note 6). In contrast to the general pattern for ancient individuals, we could not fit Hadza and Sandawe into a simple regional clade, even after accounting for recent admixture that is probably related to incoming pastoralists and farmers (contributing a total of about 41% and about 62% ancestry for these Hadza and Sandawe individuals, respectively). In particular, both were inferred to share a lineage closest to ancient foragers from

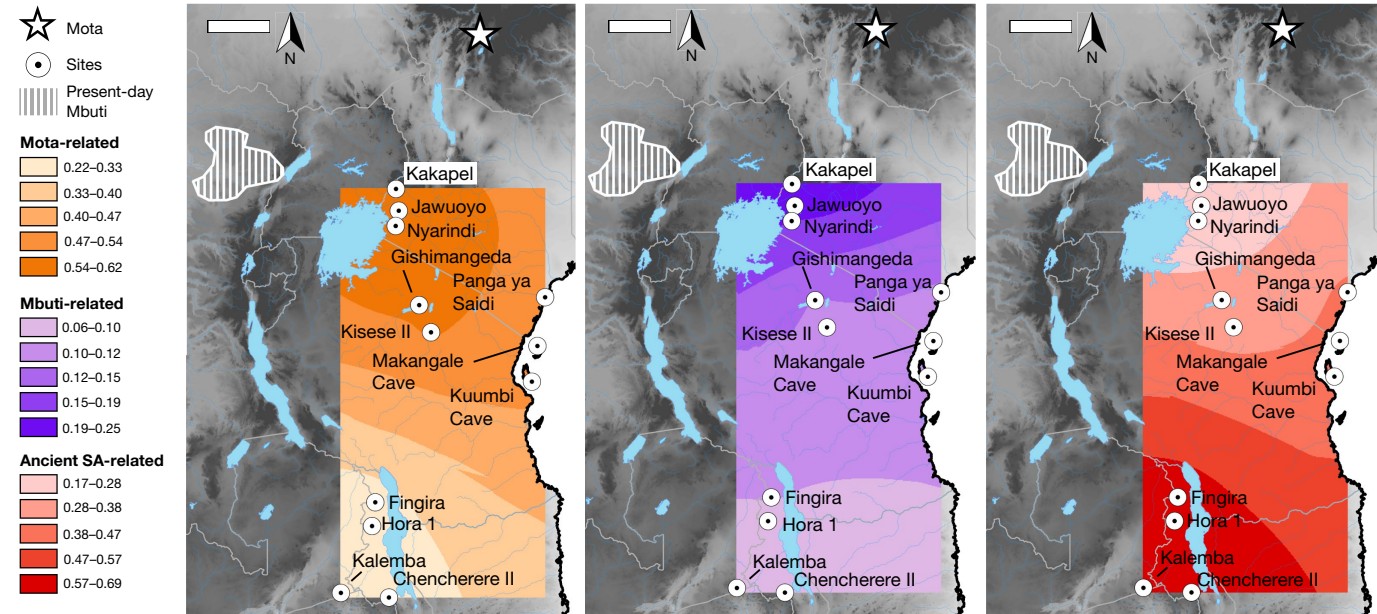

**Fig. 3 | Distribution of main ancestry components.** Kriged distribution of the proportions of each of the three main ancestry components (summing to 1) found in ancient eastern and south-central African foragers analysed in this study (details are provided in Supplementary Table 9). The approximate present-day Mbuti home region is from ref. [41]. Individuals from the same site were included using locations that differed by 0.000001 decimal degrees latitude to ensure representation in the interpolation. Scale bars, 250 km. Topographical data are from the Shuttle Radar Topography Mission (SRTM)[42]. SA, southern African.

north-central Tanzania, but the Hadza had excess allele sharing with the Mota individual, while the Sandawe had excess allele sharing with southern African foragers.

## Effective population sizes

We inferred recent (up to about 500 years before the individual's birth) ancestral effective population sizes ($N_e$) for the higher-coverage ancient individuals by scanning for long runs of homozygosity (ROH), which are expected to be present in the genomes of individuals either from populations with small sizes or whose parents have familial relatedness (the latter resulting in especially long ROH) (Methods and Extended Data Fig. 9). The calculation of $N_e$ depends on several factors in addition to census population size; in particular, $N_e$ is a function of both population density and the distance scale of those social interactions that lead to reproduction. All of the ancient individuals are inferred to have at least one long ROH (> 4 centimorgans (cM)), consistent with broad worldwide trends towards smaller population sizes in more ancient societies[23]. However, the $N_e$ estimates vary by an order of magnitude, from individuals with minimal ROH, suggesting relatively larger population sizes (I5950 (Mota): $N_e$ = 5,470, 95% confidence interval (CI) = 1,237 to unbounded; I8821 (Kisese II): $N_e$ = 2,640, 95% CI = 881–16,424) to those with an ROH of longer than 100 cM, indicative of much smaller population sizes (for example, I8808 (Jawuoyo): $N_e$ = 377, 95% CI = 229–678). Overall, the range is similar to many African forager groups today ($N_e$, around 500–1,500)[24] and towards the low end when compared with present-day population sizes worldwide[23].

## Discussion

In contrast to previous studies, our results show that a two-way clinal model extending latitudinally from eastern to southern Africa is insufficient to explain observed patterns of genetic variation in ancient sub-Saharan African foragers. Here we demonstrate that central-African-related ancestry (closest to present-day Mbuti among sampled populations), along with Mota-related and southern African-related ancestry, was ubiquitous (in varying proportions) from southwestern Kenya to southeastern Zambia (Fig. 3), with all three components present by at least about 7 ka in Tanzania and about 16 ka in Malawi. Furthermore, when considering ancient African foragers from a wide range of time periods, ecological contexts and archaeological associations, geographical proximity remains the strongest predictor of genetic similarity[5,11]. Such a pattern may indicate that long-range migrations were rare in the terminal Pleistocene and Holocene, when these individuals lived. This hypothesis is supported by the signals in our admixture graphs of excess genetic relatedness at subregional scales but not at longer-distance scales. Although it is not possible at present to estimate when and how quickly this three-way cline emerged, it must have post-dated both the emergence of the Mota-related lineage around 80–60 ka[12,16] and, with respect to the central-African-related ancestry, the split between Aka and Mbuti less than around 50 ka[25,26].

Although the observed cline of ancestry remained stable for thousands of years, we propose that it initially arose closer to this split time than to the terminal Pleistocene, and under qualitatively different patterns of mobility and admixture than after it was established. Dispersals, interactions and extensive admixture across eastern and south-central Africa before around 16 ka are evidenced by substantial proportions of ancestry related to the Mota (Ethiopia) individual as far south as Zambia, and ancestry related to southern African foragers as far north as Kenya, in combination with a high degree of homogeneity of ancestry in each subregion after that time. If patterns of mobility and social interactions had remained consistent throughout the Late Pleistocene and Holocene, we would expect to find broad evidence of longer-range ancestry connections within eastern and south-central Africa and beyond, but we observed only two significant plausible instances among our sampled individuals (involving extra central-African-related ancestry in one individual each from Kenya and Malawi).

However, within the three-way population structure, we observed distinct regional trajectories. Individuals from Kenya and Tanzania form three clusters (western Kenya, north-central Tanzania and

coastal/island), with individuals in the same cluster showing excess allele sharing even beyond what would be expected from having similar ancestry proportions. This suggests that there is elevated gene flow within each subregion, on a distance scale estimated as approximately 0–100 km. By contrast, the only signals of elevated relatedness detected for individuals from Malawi and Zambia involve those buried at the same site, and can span 1,000–3,600 years (for example, at Fingira). This pattern is best explained by low average human dispersal/interaction distances during much of the Late Pleistocene and Holocene, with the establishment of the broad-scale ancestry cline followed by, on average, more local interactions that differed by region. We observed a similar pattern in ancient foragers from western Europe, whereas those from northern and eastern Europe show longer distance scales of relatedness. This provides genetic evidence that the average distances between where people lived and where their ancestors lived (and therefore the average distances of human movement, especially with respect to reproductive partners) differed among foragers in different regions.

Our genetic findings offer new insights on demographic processes of the Late Pleistocene to Holocene that were previously studied using bioarchaeological, archaeological and linguistic evidence. Beginning approximately 300 ka, archaeological evidence attests to the long-distance movement of materials such as obsidian, presumably facilitated by social networks[27]. Exchange intensified through the Late Pleistocene to become a hallmark of the LSA, culminating in elaborate transport networks and shared material culture traditions by the Early Holocene[1,4,28,29]. However, the extent to which people were moving with objects remains an open question. Our genetic results support a scenario in which human mobility and longer-range gene flow occurred with the development and elaboration of long-distance networks approximately 80–20 ka, contributing to the formation of a population structure that persisted over tens of thousands of years during a period when people were living locally.

Genetic evidence also adds weight to arguments for changing Late Pleistocene interaction spheres, with limited gene flow accompanying changes in behaviour and possibly linguistic boundaries. However, at this juncture, we are unable to assess hypothesized population density shifts, based on heightened evidence for symbolic expression at LSA sites and the appearance and disappearance of specific artefact types[8,9,30–32]. Our genetic estimates of recent effective population size are consistent with those of at least some present-day African foragers[24], but they are not good comparators due to demographic pressures recently placed on such groups[33]. Furthermore, small subpopulations with limited gene flow could result in low ancestral effective population sizes even if the region's total population is high. Preservation of genetic diversity through the existence of many subpopulations over long time scales could also be a contributor to the high levels of genetic diversity observed in most present-day sub-Saharan African groups.

The LSA archaeological record testifies to the appearance of well-defined, temporally and spatially bounded material culture traditions[34,35], a phenomenon that is sometimes referred to as regionalization. Faunal data indicate subsistence intensification after around 20 ka[36,37], and linguistic data also suggest shifts toward local interactions, reflected in the fact that, today, communities that are presently or historically associated with foraging in central, eastern and southern Africa speak languages of different families (in central Africa, adopted from recent arrivals). At the same time, past regional connectivity and borrowing was such that linguists previously characterized 'click' languages as a single family, and the proposed grouping of Khoe–Kwadi–Sandawe strengthens evidence for longer-distance ties between eastern and southern Africa[38,39]. Our genetic results confirm that trends toward regionalization extended to human population structure, suggesting that decreasing gene flow accompanied changes in behaviour and possibly language.

## Conclusions

Demographic transformations in the past approximately 5,000 years have fundamentally altered regional population structures and largely erased what was, by the Late Pleistocene, a well-established three-way cline of eastern-, southern- and central-African-related ancestry that extended across eastern and south-central Africa. Groups who historically forage have frequently been pushed to marginal environments and have experienced transformative demographic changes, making it difficult to learn about deep history from present-day DNA. Today, Africa houses the greatest human genetic diversity, but undersampling of both living and ancient individuals obscures the origins of this diversity[40]. We show that aDNA from tropical Africa can survive from the Pleistocene and reveal patterns that could not be inferred from populations that lived even a few millennia later, underscoring the breadth of African genetic diversity and the importance of eastern and south-central Africa as long-term reservoirs of human interaction and innovation.

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

¹Department of Genetics, Harvard Medical School, Boston, MA, USA. ²Department of Human Evolutionary Biology, Harvard University, Cambridge, MA, USA. ³Department of Anthropology, University of Alberta, Edmonton, Alberta, Canada. ⁴Department of Anthropology, Stony Brook University, Stony Brook, NY, USA. ⁵Department of Anthropology and Peabody Museum of Natural History, Yale University, New Haven, CT, USA. ⁶Institute of Human Origins, School of Human Evolution and Social Change, Arizona State University, Tempe, AZ, USA. ⁷Department of Biomolecular Engineering, University of California, Santa Cruz, Santa Cruz, CA, USA. ⁸Department of Anthropology, University of Connecticut, Storrs, CT, USA. ⁹Department of Anthropology, Harvard University, Cambridge, MA, USA. ¹⁰Human Origins Program, National Museum of Natural History, Smithsonian Institution, Washington, DC, USA. ¹¹Department of History, Georgetown University, Washington, DC, USA. ¹²BIOMICs Research Group, University of the Basque Country UPV/EHU, Vitoria-Gasteiz, Spain. ¹³Department of Anthropology, University of Illinois Urbana-Champaign, Urbana, IL, USA. ¹⁴Department of Anthropology, University of South Florida, St Petersburg, FL, USA. ¹⁵School of Health Sciences, Jaramogi Oginga Odinga University of Science and Technology, Bondo, Kenya. ¹⁶Department of Anthropology, University of Oklahoma, Norman, OK, USA. ¹⁷Institutes of Energy and the Environment, Pennsylvania State University, University Park, PA, USA. ¹⁸Anthropology Program, California State University—Channel Islands, Camarillo, CA, USA. ¹⁹Independent researcher, New Haven, CT, USA. ²⁰National Museums of Tanzania, Dar es Salaam, Tanzania. ²¹Sol Solutions LLC, Scottsdale, AZ, USA. ²²Malawi Department of Museums and Monuments, Lilongwe, Malawi. ²³Department of Anthropology, Rice University, Houston, TX, USA. ²⁴Livingstone Museum, Livingstone, Zambia. ²⁵Department of Integrative Anatomical Sciences, University of Southern California, Los Angeles, CA, USA. ²⁶Department of Archaeology and Heritage Studies, University of Dar es Salaam, Dar es Salaam, Tanzania. ²⁷Department of Human Biology, University of Cape Town, Cape Town, South Africa. ²⁸Mzuzu University, Mzuzu, Malawi. ²⁹University of Malawi, Zomba, Malawi. ³⁰Department of Earth Sciences, National Museums of Kenya, Nairobi, Kenya. ³¹Interdisciplinary Center for Archaeology and Evolution of Human Behaviour (ICArEHB), FCHS, Universidade do Algarve, Faro, Portugal. ³²Department of Archaeology, Conservation and History, University of Oslo, Oslo, Norway. ³³State Key Laboratory of Loess and Quaternary Geology, Institute of Earth Environment, Chinese Academy of Sciences, Xian, China. ³⁴School of Human Evolution and Social Change, Arizona State University, Tempe, AZ, USA. ³⁵Department of Evolutionary Anthropology, University of Vienna, Vienna, Austria. ³⁶Human Evolution and Archaeological Sciences – HEAS, University of Vienna, Vienna, Austria. ³⁷Department of Anthropology, University of California, Santa Barbara, CA, USA. ³⁸Broad Institute of Harvard and MIT, Cambridge, MA, USA. ³⁹Howard Hughes Medical Institute, Harvard Medical School, Boston, MA, USA. ⁴⁰These authors contributed equally: Mark Lipson, Elizabeth A. Sawchuk. ✉e-mail: mlipson@genetics.med.harvard.edu; esawchuk@ualberta.ca; jessica.thompson@yale.edu; reich@genetics.med.harvard.edu; mary@rice.edu

## Methods

### Skeletal samples

The skeletal remains that were sampled in this study are curated at the National Museum of Kenya (Kisese II), the National Museum of Tanzania (Mlambalasi), the Malawi Department of Museums and Monuments (Hora 1 and Fingira) and the Livingstone Museum (Kalemba), and sampling permissions and protocols are described in Supplementary Note 3. Individuals were chosen based on their associated LSA archaeological contexts, and skeletal samples were selected to maximize the likelihood of yielding authentic aDNA and to minimize damage. The Fingira phalanx was an isolated find from a mixed excavation context, and too small to provide both aDNA and a direct date. A list of both successful and failing samples is provided in Supplementary Table 1. Direct radiocarbon dating was attempted on five of the six successful individuals at the Pennsylvania State University Radiocarbon Laboratory using established methods and quality control measures for collagen purification[43,44] before accelerator mass spectrometry analysis (Supplementary Note 4). A list of direct date and stable isotopic results for the two successfully dated individuals, and indirect dates where available for the other individuals, is provided in Supplementary Tables 3 and 4. All dates were calibrated using OxCal (v.4.4)[45], with a uniform prior (U(0,100)) to model a mixture of two curves: IntCal20 (ref. [46]) and SHCal20 (ref. [47]).

### aDNA laboratory work

We successfully generated genome-wide aDNA data from a total of six human skeletal elements: five petrous bones and one phalanx. We processed an additional six petrous bones, eight teeth and 11 other bones in the same manner but did not obtain usable DNA (Supplementary Table 1). In clean room facilities at Harvard Medical School, we cleaned the outer surfaces of the samples and then sandblasted (petrous bones)[48] or drilled (other bones and teeth) to obtain powder (additional information for the 15 previously published samples reported here with increased coverage can be found in refs. [11,13,15,16]). We extracted DNA[49–51] and prepared barcoded sequencing libraries (between one and six libraries for the six newly reported individuals, and between one and eight additional libraries for the previously reported individuals: from Mota Cave in Ethiopia[15] (I5950); White Rock Point in Kenya[13] (I8930); Gishimangeda Cave in Tanzania[13] (I13763, I13982 and I13983); Chencherere II (I4421 and I4422), Fingira (I4426, I4427 and I4468) and Hora 1 (I2967) in Malawi[11]; and Shum Laka in Cameroon[16] (I10871, I10872, I10873 and I10874), treating in almost all cases with uracil-DNA-glycosylase (UDG) to reduce aDNA damage artefacts[52–54]. We used two rounds of targeted in-solution hybridization to enrich the libraries for molecules from the mitochondrial genome and overlapping a set of around 1.2 million nuclear SNPs[55–58] and sequenced in pools on the Illumina NextSeq 500 and HiSeqX10 machines with 76 bp or 101 bp paired-end reads. Further details on each library are provided in Supplementary Table 2. For the Mota individual (I5950), we also generated whole-genome shotgun sequencing data, using the same (pre-enrichment) library, with seven lanes with 101 bp paired-end reads (on Illumina HiSeq X Ten machines) yielding approximately 26× coverage (1,176,635 sites covered from the capture SNP set).

### Bioinformatics procedures

From the raw sequencing data, we used barcode information to assign reads to the proper libraries (allowing at most one mismatch per read pair). We merged overlapping reads (at least 15 bases), trimmed barcode and adapter sequences from the ends, and mapped to the mtDNA reference genome RSRS[59] and the human reference genome hg19 using BWA (v.0.6.1)[60]. After alignment, we removed duplicate reads and reads with mapping quality less than 10 (30 for shotgun data) or with length less than 30 bases. To prepare data for analysis, we disregarded terminal bases of the reads (2 for UDG-treated libraries and 5 for untreated, to

eliminate most damage-induced errors), merged the .bam files for all libraries from each individual, and called pseudohaploid genotypes (one allele chosen at random from the reads aligning at each SNP). The high coverage for the Mota whole-genome shotgun data enabled us to call diploid genotypes; we used the procedure from ref. [26], including storing the genotypes in a fasta-style format that is easily accessible through the cascertain and cTools software. Code for bioinformatics tools and data workflows is provided at GitHub (https://github.com/DReichLab/ADNA-Tools and https://github.com/DReichLab/adna-workflow).

### Uniparental markers and authentication

We determined the genetic sex of each individual according to the ratio of DNA fragments mapping to the X and Y chromosomes[61]. We called mtDNA haplogroups using HaploGrep2 (ref. [62]), comparing informative positions to PhyloTree Build 17 (ref. [63]) (Supplementary Table 6). For four individuals (I2967, I4422, I4426 and I19528) with evidence of haplogroups that split partially but not fully along more specific lineages, we use the notation [HaploGrep2 call]/[sub-clade direction] (for example, L0f/L0f3 for a split on the lineage leading to L0f3 but not within L0f3). For males, we called Y-chromosome haplogroups by comparing their derived mutations with the Y-chromosome phylogeny provided by YFull (https://yfull.com).

We evaluated the authenticity of the data first by measuring the rate of characteristic aDNA damage-induced errors at the ends of sequenced molecules. We next searched directly for possible contamination by examining (1) the X/Y ratio mentioned above (in case of contamination by sequences from the opposite sex), (2) the consistency of mtDNA-mapped sequences with the haplogroup call for each individual[64] and (3) the heterozygosity rate at variable sites on the X chromosome (for males only)[65]. Two individuals (I2966 from Hora 1 and I13763 from Gishimangeda Cave) had non-negligible evidence of contamination from these metrics and also displayed excess allele sharing with non-Africans in the admixture graph analysis; we were able to fit them in the final model after allowing 'artificial' admixture from a European-related source (6% and 9%, respectively). We also restricted ourselves to damaged reads in making the mtDNA haplogroup call for I2966. Further details are provided in Supplementary Table 2 and Supplementary Note 5.

### Familial relatives

We searched for close family relatives by computing, for each pair of individuals, the proportion of matching alleles (from all targeted SNPs) when sampling one read at random per site from each. We then compared these proportions to the rates when sampling two alleles from the same individual—mismatches are expected to be twice as common for unrelated individuals as for within-individual comparisons, with family relatives intermediate. We found one possible instance between the two individuals from White Rock Point (approximately second-degree relatives, but uncertain due to low coverage) (Extended Data Fig. 1b)

### Dataset for genome-wide analyses

We merged our newly generated data with published data from ancient and present-day individuals[11–14,16,25,26,66,67]. We performed our genome-wide analyses using the set of autosomal SNPs from our target enrichment (about 1.1 million).

### PCA

We performed a supervised PCA using the smartpca software[68], using three populations (Ju|'hoansi, Mbuti and Dinka; four individuals each, from ref. [26], were chosen to create a broad separation in the PCA between highly divergent ancestral lineages from southern, central and eastern Africa) to define a two-dimensional plane of variation, and projected all other present-day and ancient individuals (using the lsqproject and shrinkmode options). This procedure captures the genetic structure

of the projected individuals in relation to the groups used to create the axes, reducing the effects of population-specific genetic drift in determining the positions of the individuals shown in the plot, as well as bias due to missing data for the ancient individuals.

## f-statistics

We computed $f$-statistics in ADMIXTOOLS[69], with standard errors estimated by block jackknife. To facilitate the use of low-coverage data, we used a new program, qpfstats (included as part of the ADMIXTOOLS package), together with the option 'allsnps: YES,' for both stand-alone $f_4$-statistics and statistics for use in qpWave and qpGraph (see below). In brief, qpfstats solves a system of equations based on $f$-statistic identities to enable the estimation of a consistent set of statistics while maximizing the available coverage and reducing noise in the presence of missing data; full details are provided in Supplementary Note 7. We computed statistics of the form $f_4$(Ind1, Ind2; Ref1, Ref2), where Ind1 and Ind2 are ancient individuals from Kenya, Tanzania or Malawi/Zambia, and Ref1 and Ref2 are either ancient southern African foragers (AncSA, listed in Extended Data Table 1), the Mota individual or present-day Mbuti. These groups were chosen in light of our PCA results and the previous evidence for ancestry related to some or all of them among ancient eastern and south-central African foragers[5,11,14].

## qpWave analysis

The qpWave software[70] estimates how many distinct sources of ancestry (from 1 to the size of the test set) are necessary to explain the allele-sharing relationships between the specified test populations and the outgroups (where 'distinct' means different phylogenetic split points relative to the outgroups). Each test returns results for different ranks of the allele-sharing matrix, where rank $k$ implies $k + 1$ ancestry sources. For absolute fit quality, we give the 'tail' $P$ value, where a higher value indicates a better fit. We also give 'taildiff' $P$ values as relative measures comparing consecutive rank levels, where a higher value indicates less improvement in the fit when adding another ancestry source. As our base test set, we used the 12 ancient eastern and south-central African forager individuals (3 from Kenya, 3 from Tanzania, 5 from Malawi and 1 from Zambia) from our admixture graph Model 3 who did not have evidence of either admixture from food producers or contamination. We also compared results when adding the Mota individual to the test set. As outgroups, we used Altai Neanderthal, Mota and the following eight present-day groups: Juǀ'hoansi, ǂKhomani, Mbuti, Aka, Yoruba, French, Agaw and Aari, with the last two (as well as Mota) omitted when we moved Mota to the test set.

## Dates of admixture

We inferred dates of admixture using the DATES software[21]. We used a minimum genetic distance of 0.6 cM, a maximum of 1 M and a bin size of 0.1 cM. As reference populations, we used ancient southern African foragers together with one of Mota, Dinka, Luhya, Yoruba or European-American individuals (the latter three from 1000 Genomes: LWK, YRI and CEU). The results assume an average generation interval of 28 years, and standard errors were estimated by block jackknife.

## Admixture graph fitting

We built admixture graphs using the qpGraph software in ADMIXTOOLS[69]. We chose to analyse each eastern and south-central forager individual separately rather than form subgroups (for example, by site or time period) to study both broad- and fine-scale structure (through relationships between individuals with both low and high degrees of ancestral similarity). Although such an approach was facilitated by our relatively manageable sample sizes, it also relied on the ability to compute $f$-statistics with our qpfstats methodology (further details are provided in Supplementary Note 7 and the '$f$-statistics' section above) to make use of all available SNPs for individuals with low-coverage data. For all of the models, we used the options 'outpop: NULL', 'lambdascale:

1' and 'diag: 0.0001.' We also specified larger values of the 'initmix' parameter to explore the space of graph parameters more thoroughly: 100,000, 150,000 and 200,000 for models 1–3 (and additional models built from them), respectively.

We began with a version of the admixture graph from ref.[16], to which we added three high-coverage ancient forager individuals (from Jawuoyo, Kisese II and Fingira) to create model 1. We then extended our model to more individuals. We used a procedure in which we (1) added each other ancient individual one by one to model 1 and evaluated the fit; (2) built an intermediate-size model 2 including a total of 11 geographically diverse eastern and south-central African foragers; (3) added the remaining individuals one by one to model 2; and (4) built our final Model 3 with all 18 individuals above a coverage threshold of 0.05× (Supplementary Note 6). In steps (1) and (3), as a starting point, we assumed a simple form of admixture (as in model 1) whereby all eastern and south-central African individuals derived their ancestry from exactly the same three sources (in varying proportions). If we found that an individual did not fit well when added in this manner, we noted the specific violation(s) to determine whether the likely cause(s) were excess relatedness to certain other individuals, distinct source(s) for the three-way admixture, admixture from other populations, or contamination or other artefacts. For the two individuals (one from Hora 1 and one from Gishimangeda) with evidence of appreciable contamination, we included dummy admixture events contributing non-African-related ancestry. Full details on our fitting procedures are provided in Supplementary Note 6.

## Excess relatedness analysis

To study excess relatedness between individuals after correcting for different proportions of Mota-related, central-African-related and southern-African-related ancestry, we built an admixture graph similar to our main model 3, but in which each forager individual is descended from an independent mixture of the three ancestry components, without accounting for excess shared genetic drift. We also included four additional individuals with lower coverage (three from Kenya and one from Chencherere II in Malawi), but excluded the two early individuals from Hora 1 due to their much greater time depth compared with other individuals in the model. Finally, for individuals modelled with admixture beyond the primary three sources (that is, pastoralist-related ancestry for four individuals, western-African-related ancestry for the Panga ya Saidi individual and the excess central-African-related ancestry for the Kakapel individual, plus dummy admixture for contamination), we locked the relevant branch lengths and mixture proportions at their values from model 3 to prevent compensation for the inaccuracies in the model by these parameters. We next used the residuals (fitted minus observed values) of each outgroup $f_3$-statistic $f_3$(Neanderthal; X, Y) to quantify the excess relatedness between individuals X and Y that is unaccounted for by the model. In other words, we fit each individual as we did during the add-one phase of the main admixture graph inference procedure (except here all simultaneously) but now, instead of using the model violations to inform the building of a well-fitting model, we used them directly as the output of the analysis.

We plotted the excess relatedness residuals for each pair of individuals as a function of great-circle distance between sites, as computed using the haversine formula (also adding a dummy value of 0.001 km to each distance). We fit curves to the data with the functional form $1/mx$, additionally allowing for translation (full equation: $y = 1/(mx + a) + b$, where $y$ is excess relatedness, $x$ is distance, and $m$, $a$ and $b$ are fitted constants) through inverse-variance-weighted least squares. We also omitted the point corresponding to the pair of individuals from White Rock Point (Kenya) because of their evidence for close familial relatedness (see above). Finally, we computed a decay scale for the curves given by the formula $(e − 1) × a/m$ (where $e$ is Euler's number). We note that a residual (that is, $y$ axis) value of zero has no special meaning in the plots.

For Mesolithic Europe, we performed two analogous analyses, one for the western part of the continent and one for eastern and northern. In the first analysis, we selected individuals with predominantly western hunter-gatherer (WHG)-related ancestry, while in the second analysis, we selected individuals who could be modelled as admixed with WHG as well as eastern hunter-gatherer (EHG)-related ancestry (Supplementary Table 12). In both cases, we built simple admixture graph models to estimate the residuals. For western Europe, we used the Upper Palaeolithic Ust'-Ishim individual from Russia[71] as an outgroup and fit all of the test individuals as descending from a single ancestral lineage. For eastern and northern Europe, we used Ust'-Ishim as an outgroup, Mal'ta 1 from Siberia[72] for a representative of ancient northern Eurasian ancestry, Villabruna from Italy[73] for WHG, Karelia from Russia[56,58,73] for EHG (admixed with ancestry related to Mal'ta and to Villabruna) and finally the test individuals each with independent mixtures of WHG and EHG-related ancestry in varying proportions.

### Effective population size inference

We called ROH starting with counts of reads for each allele at the set of target SNPs (rather than our pseudohaploid genotype data), which we converted to normalized Phred-scaled likelihoods. We performed the calling using BCFtools/RoH[74], which is able to accommodate unphased, relatively low-coverage data (at least for calling long ROH) and does not rely on a reference haplotype panel. The method is also robust to modest rates of genotype error, such as that which could occur here as a result of aDNA damage or contamination, although we recommend some caution in interpreting the results for I2966 (Hora 1) and I0589 (Kuumbi Cave; for this analysis only, we used the version of the published data with UDG-minus libraries included, for a total of around 2× average coverage). We also note that the nature of any possible effect on the final inferences is uncertain; errors could deflate the population size estimates by breaking up ROH, but they could also break very long ROH into shorter but still long blocks, which have the strongest influence on the population size estimates. In the absence of population-level data from related groups, we specified a single default allele frequency ('--AF-dflt 0.4') and no genetic map (although we subsequently converted physical positions to genetic distances using ref. [75], which we expect to be reasonably accurate at the length scales that we are interested in). For our analyses, we retained ROH blocks with length >4 cM. In three instances, we merged blocks with a gap of <0.5 cM and at most two apparent heterozygous sites between them.

From the ROH results, we applied the maximum likelihood approach from ref. [23] to estimate recent ancestral effective population sizes ($N_e$). We used all ROH blocks of longer than 4 cM, except for three individuals (KPL001 from Kakapel in Kenya, I9028 from St Helena, South Africa, and I9133 from Faraoskop, South Africa) with high proportions of very long ROH (a sign of familial relatedness between parents—approximately at the first-cousin level in these cases—rather than of longer-term low population size), for whom we used only blocks from 4–8 cM.

We note that, even within a randomly mating population, the number and extent of ROH can vary substantially between individuals, which is reflected in the large standard errors of the $N_e$ estimates for small sample sizes. We also note that recent admixture can influence ROH (and therefore $N_e$ estimates) by making coalescence between an individual's two chromosomes less likely, but on the basis of the other results of our study, we do not expect a substantial effect for these individuals.

### Reporting summary

Further information on research design is available in the Nature Research Reporting Summary linked to this paper.

### Data availability

The aligned sequences are available through the European Nucleotide Archive under accession number PRJEB49291. Genotype data used in the analysis are available online (https://reich.hms.harvard.edu/datasets). Any other relevant data are available from the corresponding authors on reasonable request.

### Code availability

Code for the bioinformatics tools and data workflows is provided at GitHub (https://github.com/DReichLab/ADNA-Tools and https://github.com/DReichLab/adna-workflow).

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

**Acknowledgements** We thank the authorities in Kenya, Tanzania, Malawi and Zambia for permission to study these ancient individuals (Supplementary Note 3); J. Stock, A. Manica and D. Bradley for previous work on the individual from Mota Cave, Ethiopia; J. Sealy for helping with the proposal to redate the Hora 1 individual; and L. Eccles for help with radiocarbon dating. Radiocarbon work was supported by the NSF Archaeometry programme (grant no. BCS-1460369) to D.J.K. and B.J.C. Excavations leading to recovery of Kahora 1 and 2 were supported by the National Geographic Society (NGS-53412R-18 to J.C.T.), Yale University and the Hyde Family Foundations. E.A.S. acknowledges support from the Social Sciences and Humanities Research Council of Canada (fellowships 756-2017-0456, BPF 169449). M.E.P. was supported the Radcliffe Institute for Advanced Study during project development. D.R. is an Investigator of the Howard Hughes Medical Institute and was also funded by NIH grants R01-GM100233 and R01-HG012287; by John Templeton Foundation grant 61220; by a private donation from J.-F. Clin; and by the Allen Discovery Center programme, a Paul G. Allen Frontiers Group advised programme of the Paul G. Allen Family Foundation. Open access publication was made possible by The John Templeton Foundation, Yale University Council on African Studies and Rice University School of Social Sciences.

**Author contributions** M.L., E.A.S., J.C.T., D.R. and M.E.P. conceptualized the study. E.A.S., J.C.T., C.A.T., K.L.R., K.M.d.L., S.H.A., J.W.A., K.J.W.A., G.A., A.B., J.I.C.-R., M.C.C., J.D., A.O.G., A.H., P.K., M.K., A.K., M.F.L., J.L., A.Z.P.M., F.M., A.M., G.M., R.M., D.M., E.N., C.O., F.S., P.R.W., D.K.W., A.Z., F.K.M. and M.E.P. provided samples, and assembled archaeological and anthropological materials and information. B.J.C. and D.J.K. performed radiocarbon analysis. N.R., R.P., J.O. and D.R. performed aDNA laboratory and data-processing work. M.L., K.A.S., I.O., N.P. and D.R. analysed genetic data. M.L., E.A.S., J.C.T., M.E.P. and D.R. wrote the manuscript with contributions from the other authors.

**Competing interests** The authors declare no competing interests.

**Additional information**
**Correspondence and requests for materials** should be addressed to Mark Lipson, Elizabeth A. Sawchuk, Jessica C. Thompson, David Reich or Mary E. Prendergast.

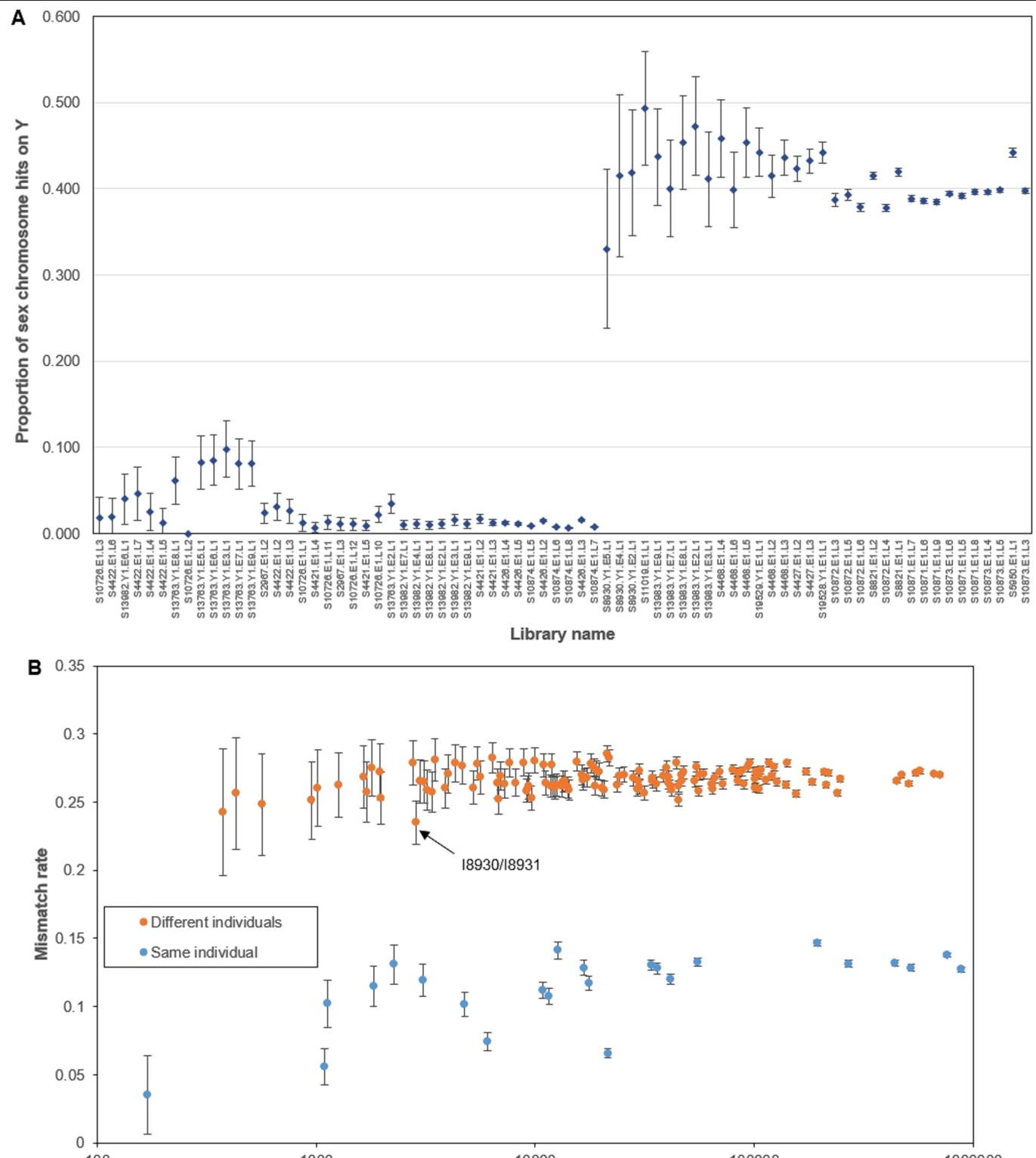

**Extended Data Fig. 1 | Sex chromosome ratios and kinship analysis. A**, Sex chromosome ratios. For each library, we show the proportion of reads aligning to chromosome Y of the total aligning to either X or Y; individuals determined to be genetically female to the left and males to the right. Bars show 95% binomial confidence intervals (normal approximation) around the mean. See also Supplementary Table 2 and Supplementary Note 5. **B**, Kinship analysis. Different-individual allelic mismatch rates (orange points) are mostly approximately twice as high as same-individual rates (blue points), as expected for unrelated individuals. The labelled pair (I8930 and I8931, both from White Rock Point), have a rate that is roughly seven-eighths that of the other pairs, which would correspond to a second-degree familial relationship (but with relatively high uncertainty given the low SNP coverage). Bars show two standard errors in each direction around the mean as determined by a Block Jackknife; note log scale for the x-axis.

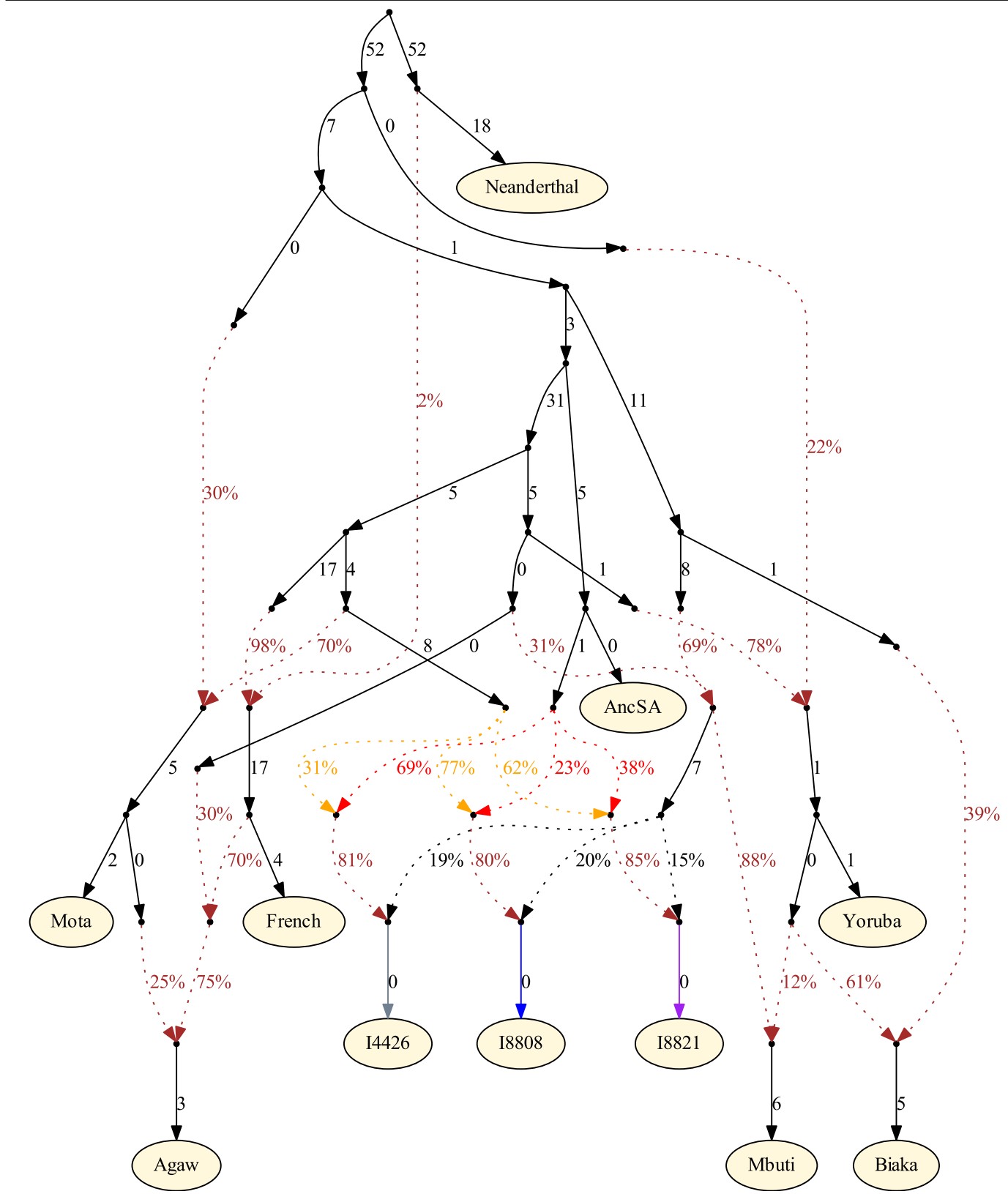

**Extended Data Fig. 2 | Full admixture graph results for Model 1.** Branch lengths are shown in units of average squared allele frequency divergence (multiplied by 1000, rounded to the nearest integer). All predicted and observed *f*-statistics agree to within Z = 2.0. AncSA = ancient southern African foragers.

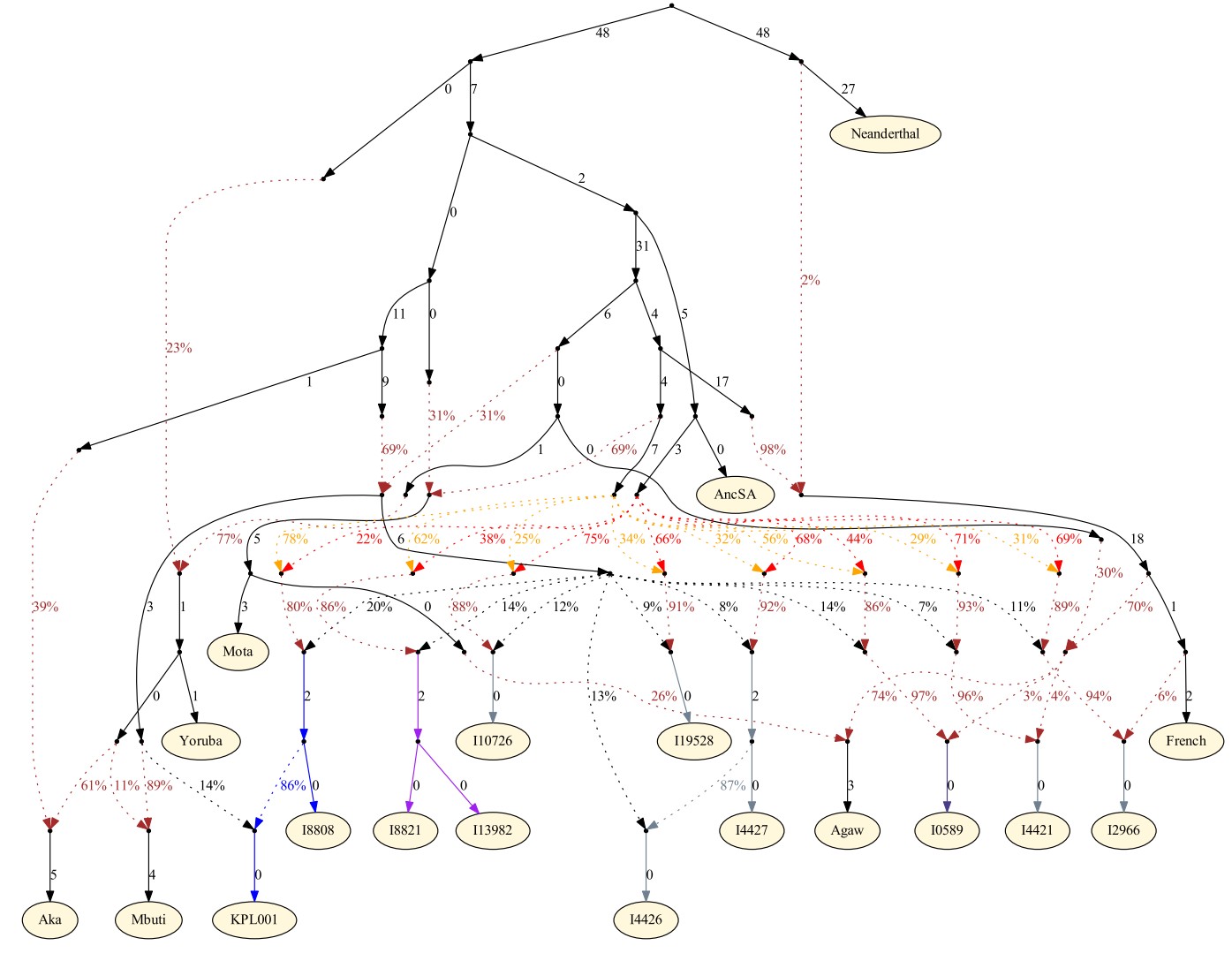

**Extended Data Fig. 3 | Full admixture graph results for Model 2.** Branch lengths are shown in units of average squared allele frequency divergence (multiplied by 1000, rounded to the nearest integer). All predicted and observed *f*-statistics agree to within Z = 3.0. AncSA = ancient southern African foragers.

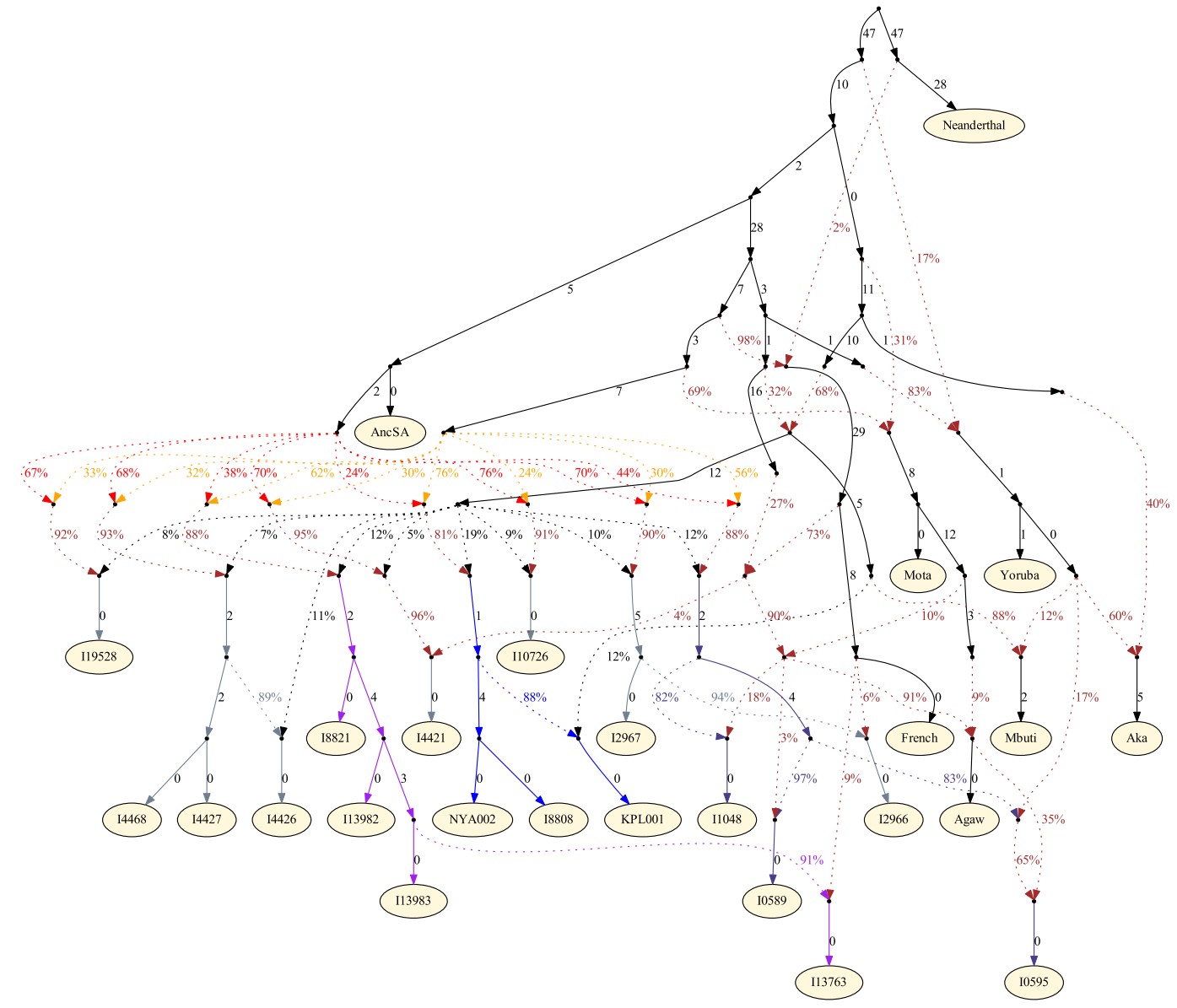

**Extended Data Fig. 4 | Full admixture graph results for Model 3.** Branch lengths are shown in units of average squared allele frequency divergence (multiplied by 1000, rounded to the nearest integer). All predicted and observed *f*-statistics agree to within Z = 3.7. AncSA = ancient southern African foragers.

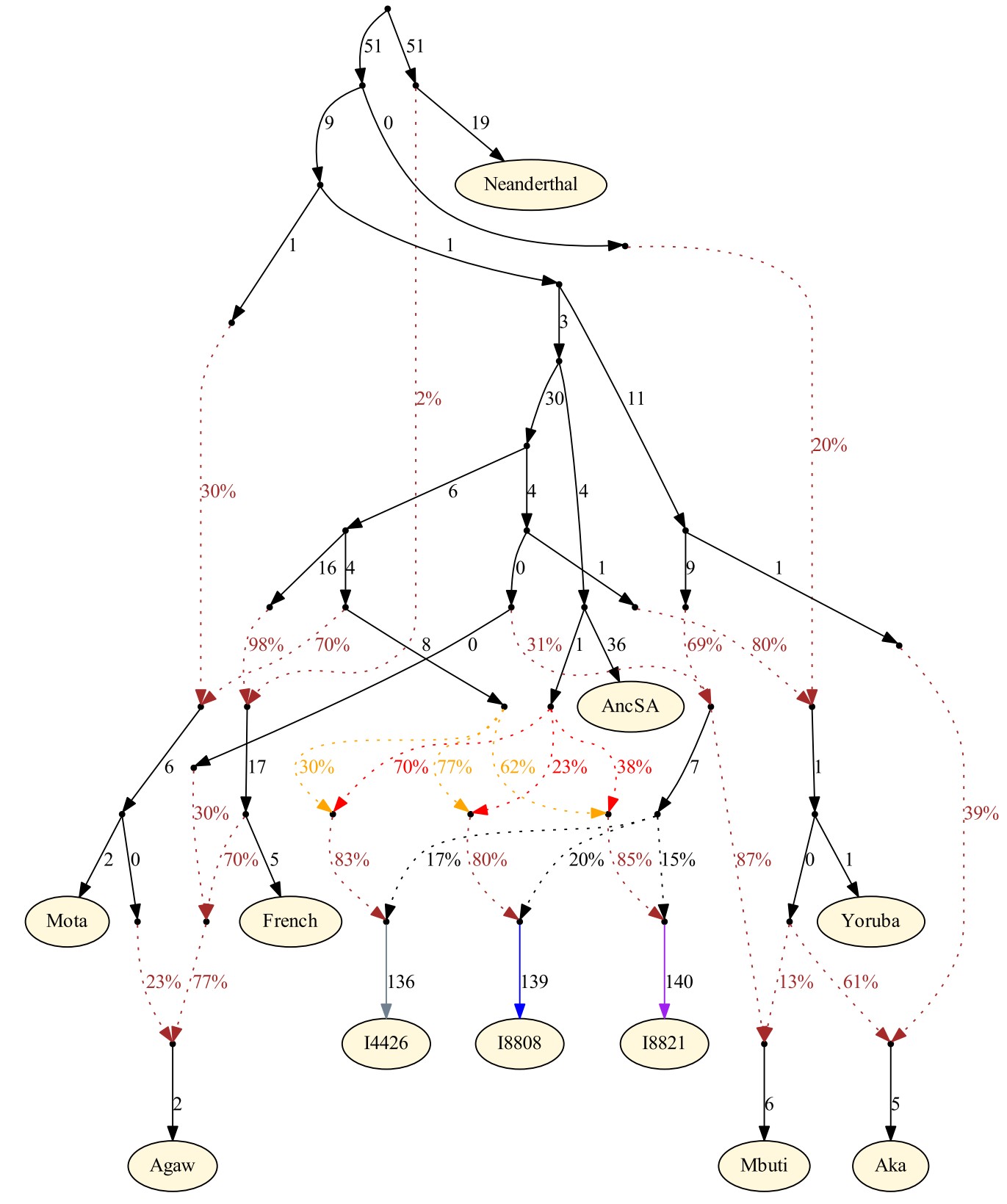

**Extended Data Fig. 5 | Admixture graph results for a version of Model 1 using only overlapping SNPs (without the *qpfstats* program).** Branch lengths are shown in units of average squared allele frequency divergence (multiplied by 1000, rounded to the nearest integer). All predicted and observed *f*-statistics agree to within Z = 2.0. AncSA = ancient southern African foragers.

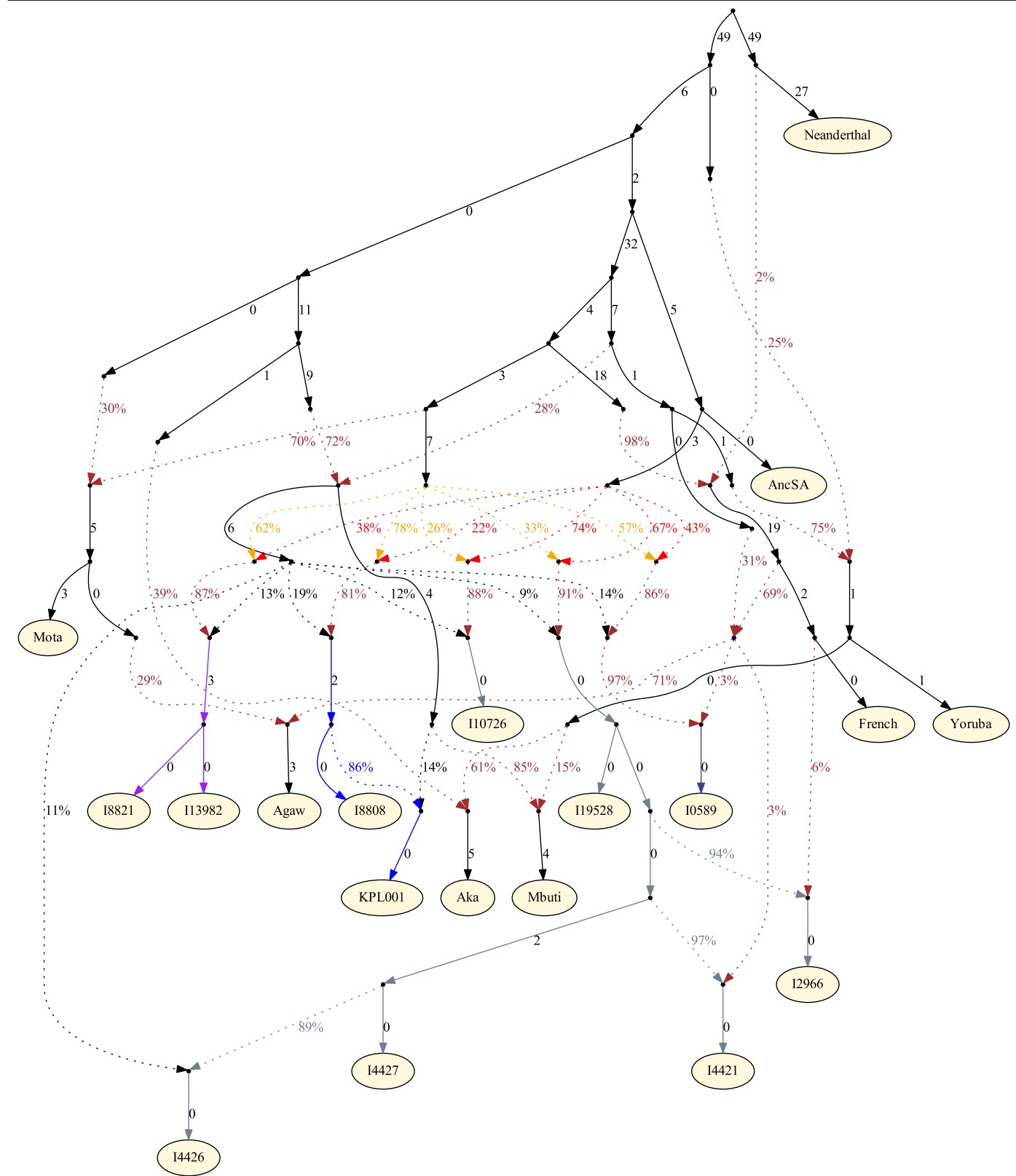

**Extended Data Fig. 6 | Admixture graph results for a version of Model 2 with the Malawi individuals fit using a shared three-way admixture clade.** Branch lengths are shown in units of average squared allele frequency divergence (multiplied by 1000, rounded to the nearest integer). All predicted and observed *f*-statistics agree to within Z = 2.9. AncSA = ancient southern African foragers.

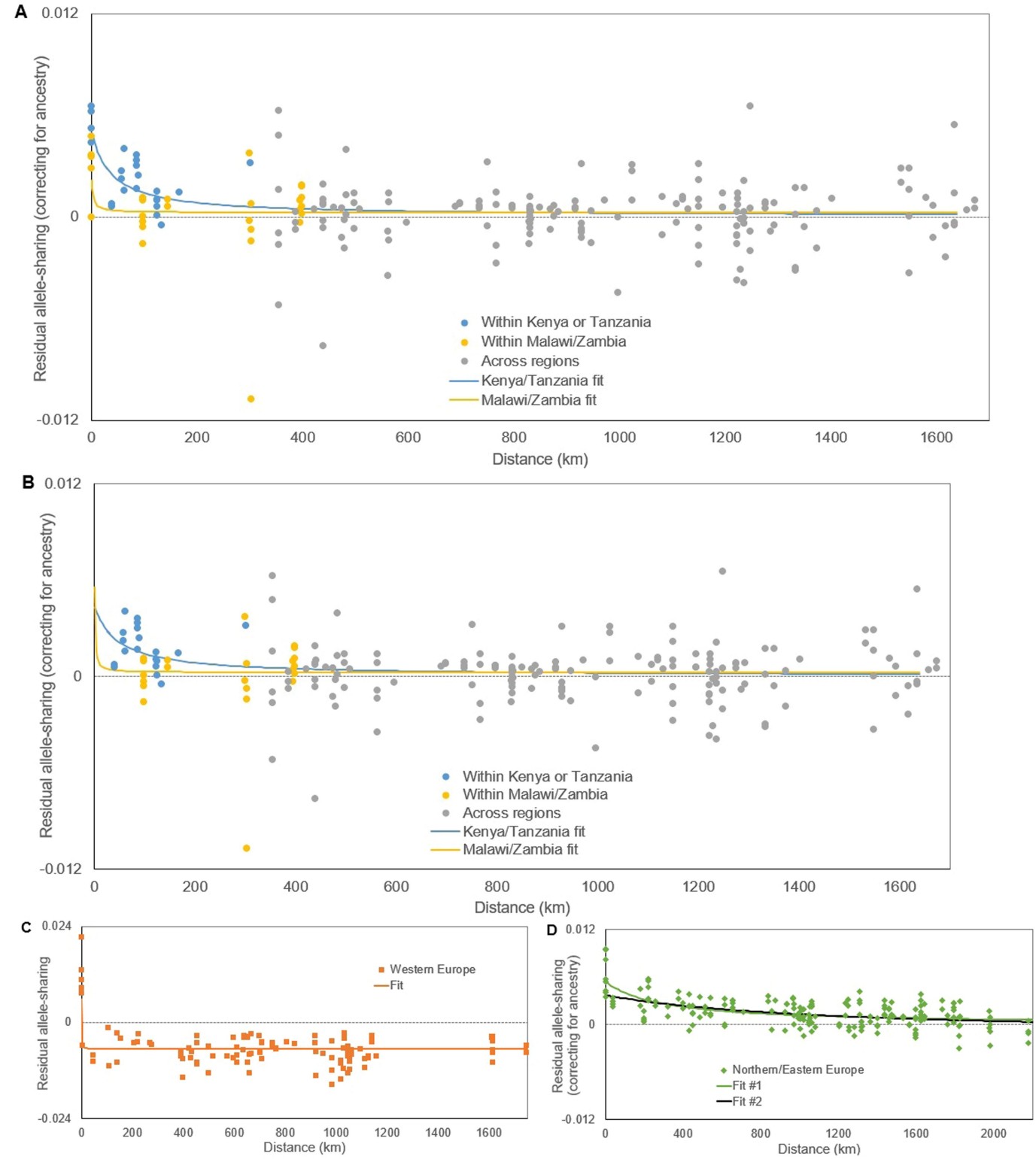

**Extended Data Fig. 7 | Excess relatedness as a function of geographical distance.** Each point represents the model residual for one pair of individuals. The lines show best-fitting curves of the functional form y = 1/kx (allowing for horizontal and vertical translation). See Methods for details. **A**, **B**: Eastern and south-central Africa, with same-site pairs omitted from the analysis in **B** (grey: different sub-regions; blue: both Kenya, both Tanzania, or both coastal; yellow: both Malawi/Zambia). **C**: Western Europe. Note the different y-axis range. **D**: Northern and eastern Europe, where fit #1 includes all pairs, while fit #2 omits same-site pairs.

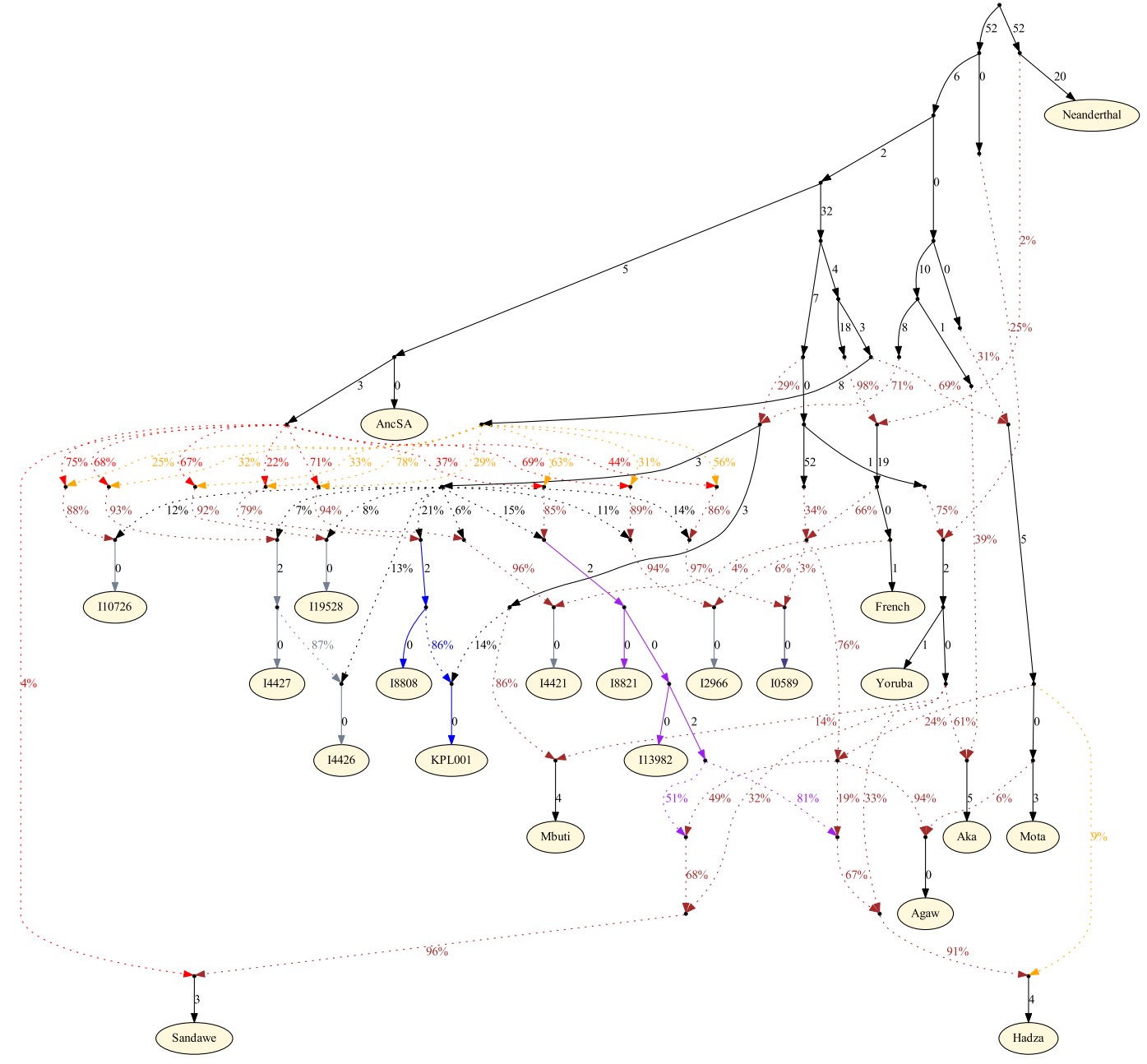

**Extended Data Fig. 8 | Admixture graph results for a version of Model 2 with Hadza and Sandawe added.** Branch lengths are shown in units of average squared allele frequency divergence (multiplied by 1000, rounded to the nearest integer). All predicted and observed *f*-statistics agree to within Z = 3.2. AncSA = ancient southern African foragers.

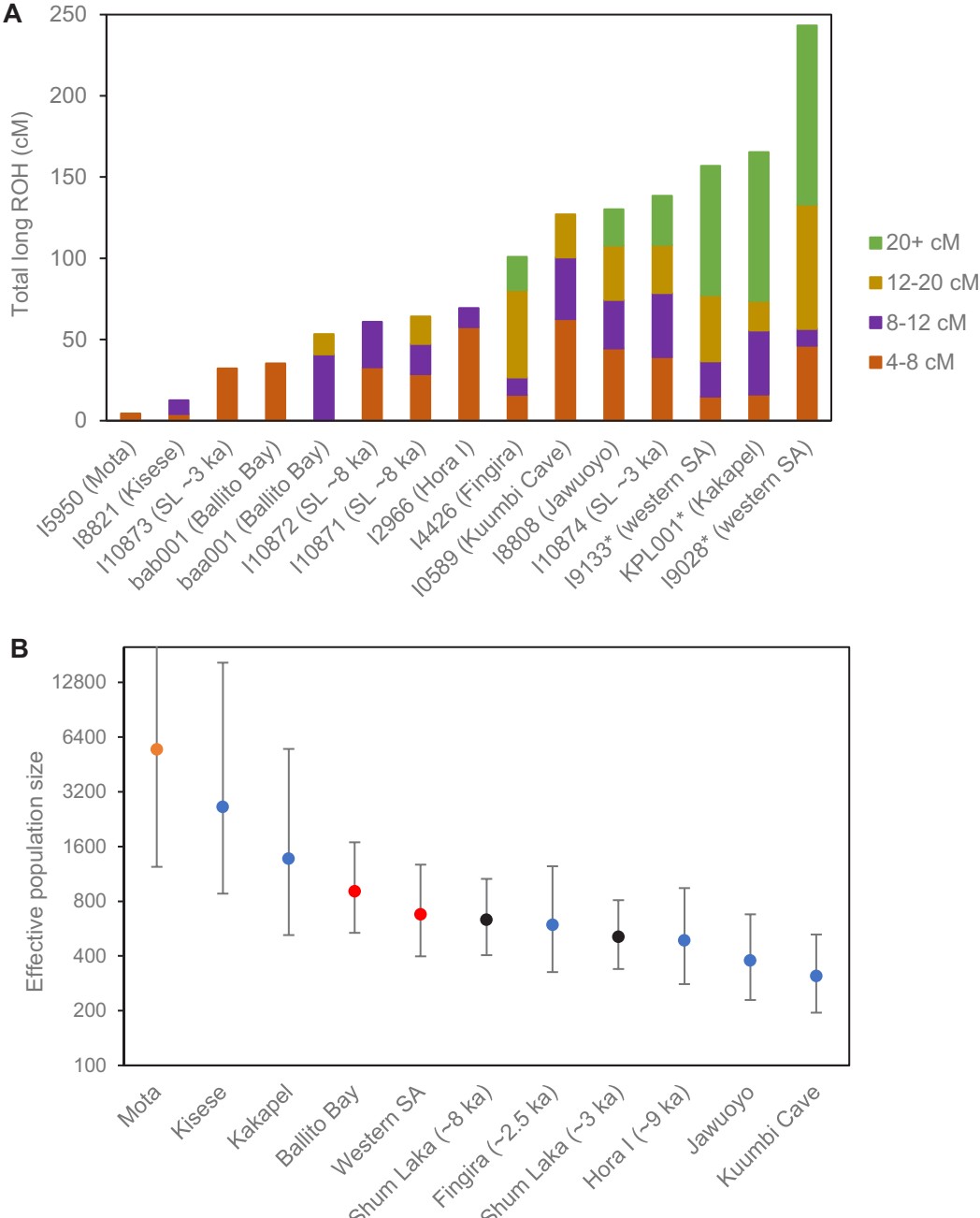

**Extended Data Fig. 9 | ROH and effective population sizes. A**: total lengths of ROH per individual in segments of > 4 cM, divided by colours into length bins. Asterisks denote individuals with evidence of familial relatedness between parents. **B**: estimated recent effective population sizes by individual or group (note log scale). Colours correspond to those in Fig. 1. Bars show 95% confidence intervals centred around the maximum likelihood estimate, reflecting uncertainty in our inferences due to the limited number in the number of ROH segments available for analysis; see Methods for details. SL, Shum Laka; western SA, western South African sites Faraoskop and St Helena.

**Extended Data Table 1 | Ancient individuals analysed in this study**

| Lab ID | Site | Context | Ele* | Sex | mtDNA hg | chrY hg | Cov† | Years BP‡ | Reference |
|---|---|---|---|---|---|---|---|---|---|
| I10871 | Shum Laka | 2/SE II | P | M | L0a2a1 | A00 | 14.139 | **7975-7795** | [16]; this study |
| I10872 | Shum Laka | 2/SE I | P | M | L0a2a1 | B | 2.644 | **7920-7700** | [16]; this study |
| I10873 | Shum Laka | 4/A | P | M | L1c2a1b | B2b | 12.963 | **3160-2970** | [16]; this study |
| I10874 | Shum Laka | 5/B | P | F | L1c2a1b | .. | 10.46 | **3210-3000** | [16]; this study |
| I5950 | Mota | Mota | P | M | L3x2a | E1b1a2b2~ | 25.686 | **4525-4300** | [15]; this study |
| KPL001 | Kakapel RS | KPL Burial 1 | P | M | L3i1 | B2b1a1~ | 0.847 | **3975-3725** | [14] |
| NYA002 | Nyarindi RS | NYA 002 | P | F | L4b2a | .. | 0.14 | **3485-3380** | [14] |
| NYA003 | Nyarindi RS | NYA 003 | P | M | ... | E | 0.016 | Failed | [14] |
| I8808 | Jawuoyo RS | Jawuoyo I | P | M | L4b2a2c | E1b1b1a1b2 | 1.369 | **1835-1740** | [13] |
| I8930 | White Rock Pt. | WRP.02.01 | P | M | L2a4 | B2b1a1~ | 0.046 | Failed | [13]; this study |
| I8931 | White Rock Pt. | WRP.03.01 | P | F | L0a2, likely | .. | 0.029 | Failed | [13] |
| I0595 | Panga ya Saidi | Burial (403) | PH | M | L4b2a2 | E1b1b1b2 | 0.136 | **500-320** | [11] |
| I1048 | Makangale | Context 204 | V | F | L0a4 | .. | 0.098 | **1505-1305** | [11] |
| I0589 | Kuumbi Cave | no. 4353 | PH | F | L4b2a2c | .. | 0.319 | **1380-1305** | [11] |
| I13763 | Gishimangeda | G-5 | P | F | L3'4 | .. | 0.093 | Failed | [13]; this study |
| I13982 | Gishimangeda | G-6 | P | F | L4b2a2 | .. | 0.254 | Failed | [13]; this study |
| I13983 | Gishimangeda | G2017.01.01 | P | M | L4b2 | B2b1a~ | 0.058 | Failed | [13]; this study |
| I8821 | Kisese II RS | KX4/5/6 | P | M | L5b2 | B2b1a~ | 3.223 | **7240-6985** | this study |
| I13976 | Mlambalasi RS | B-1 | P | F | .. | .. | 0.001 | 20,345-17,025 | this study |
| I11019 | Fingira | PF33604 | PH | M | L0d1 | B2 | 0.005 | 6179-2341 | this study |
| I4426 | Fingira | PF669 | LB | F | L0f/L0f3 | .. | 6.474 | **2675-2350; 2490-2340** | [11]; this study |
| I4427 | Fingira | PF20 | LB | M | L0d1b2b | B2b1a1~ | 0.235 | **6175-5930** | [11]; this study |
| I4468 | Fingira | PF17 | LB | M | L0d1c | B2b1 | 0.14 | **6180-5935** | [11]; this study |
| I19528 | Hora 1 | Kahora 1 | P | M | L0d3/L0d3b | B2b1a2~ | 0.177 | 16,424-14,029 | this study |
| I19529 | Hora 1 | Kahora 2 | P | M | L5b | B2b | 0.033 | 16,897-15,827 | this study |
| I2966 | Hora 1 | Hora 1 | P | M | L0k | B2b1 | 0.62 | **9090-8770** | [11]; this study |
| I2967 | Hora 1 | Hora 2 | P | F | L0a2/L0a2b | .. | 0.062 | **8175-7935** | [11]; this study |
| I4421 | Chencherere II | CHEN_SUB1 | T | F | L0k2 | .. | 0.205 | 5400-4800 | [11]; this study |
| I4422 | Chencherere II | CHEN_SUB2 | LB | F | L0k1/L0k1b | .. | 0.037 | **5050-5305** | [11]; this study |
| I10726 | Kalemba RS | SK5 | P | F | L0d1b2b | .. | 0.092 | **5285-4975** | this study |
| baa001 | Ballito Bay | Ballito Bay A | P, T | M | L0d2c1 | A1b1b2 | 12.934 | **1990-1835** | [12] |
| bab001 | Ballito Bay | Ballito Bay B | P, T, LB | M | L0d2a1 | A1b1b2 | 0.989 | **2140-1940 3320-2885** | [12,76] |
| I9133 | Faraoskop RS | UCT-386 | B | M | L0d1b2b1b | A1b1b2a | 2.328 | **2055-1750** | [11] |
| I9028 | St. Helena | UCT-606 | T | M | L0d2c1 | A1b1b2a | 1.096 | **2360-2155** | [11] |

Details for the individual corresponding to sample bab001 are available in ref. [76].

*Ele, element; B, undefined bone; LB, long bone; P, petrous; PH, phalanx; T, tooth; V, vertebra.

†Cov = coverage on genome-wide target SNPs.

‡Years BP = calibrated radiocarbon years before present. For details, see Supplementary Tables 3–4.

# Reporting Summary

## Statistics

For all statistical analyses, confirm that the following items are present in the figure legend, table legend, main text, or Methods section.

| n/a | Confirmed | |
|---|---|---|
| ☐ | ☒ | The exact sample size (*n*) for each experimental group/condition, given as a discrete number and unit of measurement |
| ☐ | ☒ | A statement on whether measurements were taken from distinct samples or whether the same sample was measured repeatedly |
| ☐ | ☒ | The statistical test(s) used AND whether they are one- or two-sided<br>*Only common tests should be described solely by name; describe more complex techniques in the Methods section.* |
| ☒ | ☐ | A description of all covariates tested |
| ☐ | ☒ | A description of any assumptions or corrections, such as tests of normality and adjustment for multiple comparisons |
| ☐ | ☒ | A full description of the statistical parameters including central tendency (e.g. means) or other basic estimates (e.g. regression coefficient) AND variation (e.g. standard deviation) or associated estimates of uncertainty (e.g. confidence intervals) |
| ☐ | ☒ | For null hypothesis testing, the test statistic (e.g. $F$, $t$, $r$) with confidence intervals, effect sizes, degrees of freedom and $P$ value noted<br>*Give P values as exact values whenever suitable.* |
| ☒ | ☐ | For Bayesian analysis, information on the choice of priors and Markov chain Monte Carlo settings |
| ☒ | ☐ | For hierarchical and complex designs, identification of the appropriate level for tests and full reporting of outcomes |
| ☒ | ☐ | Estimates of effect sizes (e.g. Cohen's *d*, Pearson's *r*), indicating how they were calculated |

*Our web collection on statistics for biologists contains articles on many of the points above.*

## Software and code

Policy information about availability of computer code

| Data collection | BWA v0.6.1, HaploGrep2 v2.1.19, cTools v11, contamMix v1.0-10, ANGSD v0.923, other bioinformatics tools and data workflows (https://github.com/DReichLab/ADNA-Tools and https://github.com/DReichLab/adna-workflow), OxCal v4.4 |
|---|---|
| Data analysis | ADMIXTOOLS v7.0.2 (includes new qpfstats software), EIGENSOFT v7.2.1, hapROH v0.3, DATES v1 |

For manuscripts utilizing custom algorithms or software that are central to the research but not yet described in published literature, software must be made available to editors and reviewers. We strongly encourage code deposition in a community repository (e.g. GitHub). See the Nature Portfolio guidelines for submitting code & software for further information.

## Data

Policy information about availability of data

All manuscripts must include a data availability statement. This statement should provide the following information, where applicable:
- Accession codes, unique identifiers, or web links for publicly available datasets
- A description of any restrictions on data availability
- For clinical datasets or third party data, please ensure that the statement adheres to our policy

The aligned sequences are available through the European Nucleotide Archive under accession number PRJEB49291. Genotype data used in analysis are available at https://reich.hms.harvard.edu/datasets. Any other relevant data are available from the corresponding authors upon reasonable request.

# Field-specific reporting

Please select the one below that is the best fit for your research. If you are not sure, read the appropriate sections before making your selection.

☐ Life sciences ☐ Behavioural & social sciences ☒ Ecological, evolutionary & environmental sciences

For a reference copy of the document with all sections, see nature.com/documents/nr-reporting-summary-flat.pdf

# Ecological, evolutionary & environmental sciences study design

All studies must disclose on these points even when the disclosure is negative.

| | |
|---|---|
| Study description | Population genetic analyses were performed on DNA data generated from ancient human skeletons as well as present-day individuals. Historical relationships were inferred primarily from allele-sharing patterns across populations, computed using genome-wide SNP genotypes. |
| Research sample | Six newly reported ancient human individuals buried in what are now Zambia, Malawi, and Tanzania; 28 previously published ancient individuals (15 with increased sequencing coverage in this study); published comparative data from present-day groups. Ancient DNA data are severely lacking from forager groups from sub-Saharan Africa, so we aimed to sample all the ancient skeletal remains associated with foraging in Later Stone Age archaeological contexts that we could access while respecting ancient DNA ethics guidelines that have the goal of preserving remains for future analysis, with the goal of filling in gaps in knowledge.  We recognize that we were not able to sample from the great majority of contexts and there remain many gaps in available ancient DNA data from Africa. |
| Sampling strategy | We tested a total of 31 skeletal samples and obtained working data from six. We targeted approximately 1.2 million genome-wide SNPs in generating DNA data, which effectively cover almost all independent loci in the genome (due to linkage disequilibrium) and provide good power in population history analyses. We generated whole genome sequencing data for individuals for which the proportion of human DNA was high enough to make this possible, namely the individual from Mota Cave, Ethiopia. |
| Data collection | DNA from the ancient remains was extracted, sequenced, and processed into SNP genotype calls in the laboratory of D.R. |
| Timing and spatial scale | Ancient individuals were sampled from archaeological sites in eastern and south-central Africa with evidence for foraging lifeways and in association with Later Stone Age material culture. We attempted to sample as comprehensively as possible within this framework but due to the limitation in sample availability there remain substantial gaps in space and time. |
| Data exclusions | As noted above, the majority of the skeletal samples did not yield working data as assessed by standard ancient DNA quality criteria. In our primary admixture graph-based analyses, we restricted to individuals with sufficient sequencing coverage (> 0.05x) to give meaningful results. |
| Reproducibility | The genetic findings were reproduced for many samples through making multiple ancient DNA libraries which yielded consistent findings about genetic relatedness patterns. |
| Randomization | Most analyses of ancient individuals were done one-by-one, but in some cases subgroups were defined based on location and/or time period, or (for present-day individuals) ethno-linguistic groups. |
| Blinding | Blinding was not possible for this study due to the importance of co-analyzing each genetic datapoint closely in concert with the archaeological data. We performed all analyses, however, based on testing the null assumption (disproved in all case) of symmetrical genetic relationship of each individual to all other individuals. |

Did the study involve field work? ☐ Yes ☒ No

# Reporting for specific materials, systems and methods

We require information from authors about some types of materials, experimental systems and methods used in many studies. Here, indicate whether each material, system or method listed is relevant to your study. If you are not sure if a list item applies to your research, read the appropriate section before selecting a response.

## Materials & experimental systems

| n/a | Involved in the study |
|---|---|
| ☒ | ☐ Antibodies |
| ☒ | ☐ Eukaryotic cell lines |
| ☐ | ☒ Palaeontology and archaeology |
| ☒ | ☐ Animals and other organisms |
| ☒ | ☐ Human research participants |
| ☒ | ☐ Clinical data |
| ☒ | ☐ Dual use research of concern |

## Methods

| n/a | Involved in the study |
|---|---|
| ☒ | ☐ ChIP-seq |
| ☒ | ☐ Flow cytometry |
| ☒ | ☐ MRI-based neuroimaging |

| | |
|---|---|
| Specimen provenance | Malawi:<br>The two Hora 1 burials were recovered in 2019 and are curated by the Malawi Department of Museums and Monuments (formerly Department of Antiquities), now under the Ministry of Youth, Sports, and Culture. Additional sampled individuals from Mazinga and Hora were recovered in 2017 and 2018. The individuals from Fingira and Mtuzi/Chencherere II were recovered during 2016 fieldwork or retrieved from the National Repository in Nguludi, respectively. Permission for the research, including both excavation and ancient DNA protocols for both failed and successful samples, was provided under permits A/III/3.3/70, A/III/3.3/71, AD/23/56, and NCST/RTT/2/6. Export was provided under A/1/1/1/3.6/50, A/1/1/1/3.6/44, A/II/1.5/33, and MHQ/CUL/1/04/2.<br><br>Zambia:<br>Skeletal remains from Kalemba Rockshelter are curated at Livingstone Museum. Permissions to conduct research and to export skeletal remains for destructive sampling were received from the Director of the Livingstone Museum and from the National Heritage Conservation Commission (Permit NHCC/8WR/004/17).<br><br>Tanzania/Kenya:<br>For Kisese II Rockshelter, the site is located in Tanzania, but burials are currently curated in National Museums of Kenya in Nairobi. Permissions for aDNA sampling obtained in Kenya from the National Commission for Science, Technology, and Innovation (NACOSTI permit P/17/34239/17088), and through affiliation with the National Museums of Kenya (NMK); and in Tanzania from the Commission for Science and Technology (COSTECH permits 2017-220/221/222-NA-2012-50). Permission to export the skeletal samples from Kenya for destructive sampling was issued by the Cabinet Secretary, Ministry of Sports and Heritage, Kenya.<br><br>For Mlambalasi Rockshelter, the sampled burials are curated at the National Museum and House of Culture in Dar es Salaam, Tanzania. Permission to sample these remains was granted by the Commission for Science and Technology (COSTECH permits 2017-220/221/222-NA-2012-50), through affiliation with the National Museums of Tanzania (NMT). Permission to export the skeletal samples for destructive sampling was granted by the Division of Antiquities, Ministry of Natural Resources and Tourism (Export License 03/2018/2019). |
| Specimen deposition | Skeletal tissue samples exported from Kenya were repatriated to the National Museum of Kenya (NMK) in Nairobi in October 2017 and June 2019. While no intact tissue samples remain outside the country, remaining powder, DNA extracts, and libraries remain under curation at the Reich Laboratory at Harvard University as agreed upon with the NMK Head of Earth Sciences. All skeletal tissue samples exported from Tanzania were repatriated to the NMT in May 2019, in keeping with a Memorandum of Agreement (MOA), which allows for curation of remaining powder, DNA extracts, and libraries in the Reich Laboratory. All skeletal tissue samples from the Livingstone Museum were repatriated to that institution in June 2018, except for tissue remaining after radiocarbon dating, which was repatriated in June 2019. No intact tissue samples remain outside of Malawi, and remaining powder, DNA extracts, and libraries remain under curation at the Reich Laboratory at Harvard University. |
| Dating methods | Three new dates are provided, one on dental enamel and two on bone collagen. All three dates were obtained at the Pennsylvania State University (PSU) Radiocarbon Laboratory, via accelerator mass spectrometry (AMS). Details of sample pretreatment and radiocarbon dating procedures are provided in Supplementary Note 3. Radiocarbon ages were calibrated using OxCal version 4.4, employing a uniform prior (U(0,100)) that allows the program to model an unspecified mixture of two curves: IntCal20 and SHCal20. |

☒ Tick this box to confirm that the raw and calibrated dates are available in the paper or in Supplementary Information.

| | |
|---|---|
| Ethics oversight | Detailed proposals including protocols for minimally destructive sampling were provided to and approved by the permit-granting and curating authorities listed above: In Malawi, the Malawi Department of Museums and Monuments (formerly Department of Antiquities); in Zambia, the Livingstone Museum and the National Heritage Conservation Commission; in Tanzania, the Commission on Science and Technology and National Museum of Tanzania; and in Kenya, the National Commission for Science, Technology, and Innovation, and the National Museums of Kenya. |

Note that full information on the approval of the study protocol must also be provided in the manuscript.

