## [Peer Review File · Nature]

Manuscript Title: Ancient DNA and deep population structure in sub-Saharan African foragers

Reviewer Comments & Author Rebuttals

Reviewer Reports on the Initial Version:

Referee #1 (Remarks to the Author):

Summary: The paper's strength is that it provides ancient DNA from the time period before 5000 years ago, i.e. preceding major demographic changes across sub-Saharan Africa. It also proposes to understand the LSA transition and a previously identified population cline that no longer exists between eastern and southern Africa. Current aDNA from sub-Saharan Africa is of a Holocene age. The results suggest that a long period of long distance exchange and population mixing in Africa was replaced by strongly structured local populations from around 20 ka onwards, until the spread of farming. These shed light on aspects of deep population history in Africa, but certain aspects of the paper either lack detail or are a little too technical for a broad reach paper.

I am not able to review the genetics components of this paper and leave this to a specialist in this subject area. However I evaluate the fit between the genetics and the archaeology and recommend some changes. With these changes, I recommend the paper for publication as an important contribution to the field.

Originality and Significance: The authors present new genome wide aDNA data with radiocarbon ages from three very Late Pleistocene and three early to mid-Holocene associated with LSA technology. Their results extend the previously identified cline both spatially and temporally, and find further complexity within it. Critically, and this is where the novelty of the paper lies, they also show that there are other sources of for the observed complexity. They show that populations ancestral to the studied individuals had significant interaction and mixing across vast region spanning the eastern coast to the interior across eastern and south-central Africa. However, this section is somewhat convoluted and a little tricky to follow. The authors sometimes explain terms and hypotheses more clearly than at other times in the paper. It would be good if they went through this entire text and tried to make it a little clearer for general readership.

What is interesting about the results is that while many archaeologists assume long-distance contact between foraging groups, these results suggest that, at least for the time period studied which post-dates ca. 80-60 ka, this was not the case across as vast region. That said, they do hypothesise that prior to the emergence of the three-way cline, long-range dispersals and significant admixture did take place at some point in the Pleistocene, likely before 50 ka and certainly before 20 ka. I would like to see some elaboration on this in the paper. I understand it is a hypothesis that might be over-stretching the data a bit, but I think it is nevertheless important to set up more comprehensively. These populations became strongly structured again around 20 ka, remaining strongly local until the expansion of farming in Africa. I would like to see a little more elaboration here on key features of the LSA that support this argument. Currently it is a little light on detail and generic, given the paper starts off with saying it is shedding light on the LSA transition.

Data and Methods: The SI details on excavation and sampling are clear and well written. The radiocarbon ages are not all what could be described as precise and some individuals could be younger than stated, which is acknowledged by the authors. Other ages are indirect. For the oldest individual, where a date of 18.5ka is given in the main text, this is really a maximum age and I note there is a stratigraphic inversion at the site. This individual too, could be a little younger. Nevertheless, the authors present convincing arguments for the approximate ages given, and any deviations from the estimates are unlikely to exceed a couple of thousand years at most for the

Pleistocene, and I doubt more than about one thousand for the Holocene. I would like to see the main paper make clear which age estimates were maximum estimates, and couple these with minimum estimates to give the reader a better understanding of these age brackets, particular when so many of the dates are problematic.

Conclusions: I think the authors do a good job of not over-stretching their results and make clear what is a hypotheses requiring more work, and what is well supported by the data they offer. I cannot comment on the genetics-specific analytical work and its quality.

References: No doubt a few further archaeological references will be added with the recommended revisions, however the references seem adequate.

Clarity/Context: In the section on uniparental markers, please explain Y chromosome haplogroups as you do for mtDNA haplogroups. What is the significance of chrY haplogroup B2? The text talking about 'calling' Y chromosome, etc. also reads oddly to the general reader. If this is a technical term, I recommend changing it to something more intuitive for Nature's general readership.

More generally, it would be helpful if some technical terms such as 'excess allele sharing' were explained since the paper is for a broad readership across different disciplines. As mentioned above, it would be helpful if the methods and results sections were a little clearer and less technical. It would be really helpful if there was a brief summary in there about all the population groups the authors speak about and what is known about them. Currently information is a little distributed within the text. Please also check your estimates for the beginning of the LSA, and this is important. They do not match the evidence from Panga Ya Saidi.

Referee #2 (Remarks to the Author):

In order to investigate the population history of Late Pleistocene/Holocene African foragers, Lipson et al. generated and analyzed genome-wide data of six completely new ancient modern humans from sub-Saharan Africa and improved the quality of data of 16 previously published ancient individuals (most notably the Mota individual for who they now report genome wide data to 25x coverage). The data generated represent an exciting contribution which should be of interest for a wide range of researchers in the field of human population genetics and evolution. Moreover, by successfully recovering ancient DNA from 3 Late Pleistocene individuals they are doubling the time depth of ancient human DNA reported from sub-Saharan Africa. The study demonstrates well that although the conditions of ancient DNA preservation in Africa are not optimal (about 1/5th of the specimens screened show evidence of ancient DNA preservation, and median coverage is 0.06x), there are at least some sites and remains that can be made to work well enough.

However, the study doesn't really provide substantial new outcomes that would change the way we think about ancient foragers populations in Africa in the late Pleistocene/Holocene. Basically, it replicates and investigates further previous findings that showed a gradient of relatedness between eastern and southern African foragers during the Holocene, highlights the presence of central African-related ancestry in ancient eastern and south-central African foragers and provides evidence that this population structure existed already during the Late Pleistocene as early as 18 kya. The study shows that genetic similarity among ancient eastern African foragers correlates well with geographic proximity. The authors also calculate effective population sizes and show that these are in the range of what is known for African forager groups today.

General major comments

1. The inconsistency in the way the individuals are referred in the main text, figures, legends and supplementary materials (sometimes by geographic location, sometimes archaeological name, sometimes by Individual ID or by different combinations of these) makes the reading of the

manuscript unpleasant and difficult to follow. In order to help readers interpret the results without having to keep looking up to relevant informations about the individuals in Table 1, I advise the authors to adopt a nomenclature similar to the one used in Skoglund et al. 2017 or to consistently give the estimated ages and geographic locations of individuals at relevant places in the text as for example for samples discussed between lines 177 and 189.

2. Both mtDNA and nuclear sequences support the conclusion that individuals from each region sampled cluster with one another with few exceptions. However, the authors barely comment on specific individuals/populations context and whether there is an age-related signal here. Considering the age range of the samples (~20,345-17,025 cal BP to 500,320 cal BP), I advise the authors to integrate more systematically the time scale in their discussions.

3. The range of values and consistency between methods of contamination estimates, as well as the potential effect of contamination on all results is not presented or discussed enough in the main text. While the authors reported that they estimated contamination using 3 different approaches, only the results of the heterozygosity rate at variable sites on the X chromosome (for males only) are presented Table S1. I would like to see the actual contamination estimates from each method for the samples. eg: What means "small amount of contamination" (Line 362) or "a non-negligible amount of contamination" (Line 924)? In the qpGraph analyses, two samples are modelled post-hoc as contaminated by present-day European DNA only in cases where a lack of fit lead the authors to attempt to include contamination in their estimates. What about potential effects of contamination on the PCA and f-statistics? I'm also worry about that possible contamination by present-day Africans may have gone unnoticed. The study requires additional estimates of contamination using existing approaches such as those based on linkage (Nakatsuka et al. 2020) or on damage (Meyer et al. 2016; Peyrégne and Peter 2020).

4. The statistics used in the study are state-of-the-art methods and are all appropriate. However, the description of uncertainties and probabilities values should be better reported in the figures or the figure legends. For example, for the PCA analysis it would be nice to report in the figure or the legend how much genetic variation each PC captures. Figure 1C doesn't have axes, which makes it very confusing at first. For the qpGraphs, the Z-score of the lowest/highest outlier should be reported in the figure or the legend.

Specific major comments

1. The analysis of 'phenotypic SNPs' presented line 193-195 should be a bit more detailed, as written it is not sure what message should the reader take home here? Are the data not good enough in any individual to know whether these alleles were present? Or were these mutations not yet present in any of these individuals. Given that there are some individuals with good coverage (e.g. Mota, who hasn't been included for these analyses according to Table S6) there must be some of the individuals where one can say for sure whether they carried these variants or not?

2. Figure 2 is comprehensive only after reading the whole main text and supplementary Note 4. The legend should give more details to help the reader to understand it, especially the lower part. I suggest to the authors to clarify how they selected the different cluster, to include the proportion of the three-way admixture clade at the top of each cluster sub-graph and to indicate the lengths of the different branches in the sub-graphs. Indication of the age of the individuals should also help the readers to better interpret the results.

3. The qpGraph analyses supporting the inter and intra-regional relationship presented seem robust and were done intelligently, with a lot of discernment and care. The graph of Model 3 Figure S4 is particularly impressive and certainly required a lot of work. However, they require also a lot of time and focus to decipher, and most admixture events are impossible to trace visually in Model 2 and 3 (Figure S3, S4 and S6). I think it will be easier to the readers to follow if the admixture analyses are presented in a more gradually manner. Following the strategy used to make the

graph Figure S6, I suggest to the author to make independent graphs for each of the different cluster they identified (Kenya, Tanzania, Coastal/Island and Malawi/Zambia), to discuss them separately before putting the complete picture of the Model 3 graph together. This should help to better visualize and comprehend the contrast in structure between the Malawi/Zambia cluster and the Tanzania/Kenya cluster. Also, the authors should clarify the choice of individuals used in the different graphs, for example why individuals from Malawi I4468 and I2967 are not included in the graph of the shared three-way admixture clade of the Malawi individuals presented figure S6? Using different colors for the different cluster of individuals should make the graph easier to read.

General minor comments

1. Although the number of genomes that are now available from late Pleistocene and Holocene African individuals is growing, the number of individuals is still small and the sampling density in time and space uneven and low. Tanzania and Kenya are referred to as East Africa and Malaysia and East Zambia as South-Central Africa. The authors should probably be a little more cautious about what insights into 'continental scale' population structure can be detected.
2. The authors successfully modelled the ancestries in ancient Eastern African foragers as deriving from three divergent source population and that there may possibly be an additional source contributing to the Mota individual from Ethiopia. It might be good for the authors to discuss the constraints here ; in regards of the limited sampling and reference set as well as the data quality, inferences can only be made about *these* foragers. The conclusion seems unlikely that there is only these three ancestry sources for all these populations (as illustrated by the Mota result). The idea that there may exist of as-yet unsampled populations that contribute to later Southern and Northeastern foragers is very interesting.

Specific minor comments

1. The introduction feels a bit flat and do not emphasized the exciting specific questions and previous findings related to prehistoric African population history, to the exception of the genetic cline between eastern and southern African foragers. The first paragraph lacks references and can be skipped since the most relevant information in it is repeated in the third paragraph. The second paragraph introduces in broad details the transition to the Later Stone age (LSA), for which I hardly see the relevance in regard to the main results of the study. I think it can be shortened.
2. The author inverted and rotated the PCA plot presented in figure 1C, I assume in order to match the geographic distribution presented in Figure 1A. I found it very confusing, and I haven't understood at first what was the motivations. I suggest figure 1B and 1C to be presented in the same orientation, whether it is inverted and rotated or not.
3. The present-day individuals/populations used to compute the principal component of the PCA analysis are not reported in the plot, I am curious to see how they positioned compared to the ancient individuals.
4. The unit for related ancestry found in ancient African foragers in figure 5 is not reported.
5. The conclusion is a bit bland. I think the authors should rewrite it, they should remove the summary of the discussion and emphasize better the context and the relevance of their results. They should give more specific perspectives about the future of ancient DNA for the study of prehistoric Africa.
6. Supplementary Note 1 and 3 are inverted.
7. The archaeological site summaries part in Supplementary Note 2 is very inconsistent between sites. For Fingira and Hora 1 sites the description focus more on the archaeological context and the

dating of the samples while for Kalemba, Kisese II and Mlambalasi rockshelters we have in additional information about the site. I suggest changing the title "Archaeological site summaries" of that section of Supplementary Note 2 to "Samples archaeological context and dating", as I'm afraid part of the audience would skip that section thinking it is only about the sites.

8. The authors should clarify the relevance of the intermediate admixture graph Model 2.

9. In Supplementary Note 4, in order to explain their expanded models, the authors described intermediate qpgraphs which are not presented anywhere (e.g. lines 679 to 689, 691 to 700 or 825 to 835). It is very difficult to apprehend the graph only from writing description. I suggest to the authors to present every intermediate graphs described.

10. In Supplementary Note 4, I think the section "Relationships of the three primary forager ancestry sources to other sampled populations" should be presented before the section "Detailed modeling of forager individuals". It is more intuitive to read first about the big picture before going to details.

References:

1. Skoglund, P. et al. Reconstructing Prehistoric African Population Structure. *Cell* 171, 59-71.e21 (2017).
2. Nakatsuka, N. et al. ContamLD: estimation of ancient nuclear DNA contamination using breakdown of linkage disequilibrium. *Genome Biol.* 21, 199 (2020).
3. Peyrégne, S. & Peter, B. M. AuthentiCT: A model of ancient DNA damage to estimate the proportion of present-day DNA contamination. *Genome Biol.* 21, (2020).
4. Meyer, M. et al. Nuclear DNA sequences from the Middle Pleistocene Sima de los Huesos hominins. *Nature* 531, 504-507 (2016).

Author Rebuttals to Initial Comments:

Referees' comments: Authors' responses in blue font

Referee #1 (Remarks to the Author):

Summary: The paper's strength is that it provides ancient DNA from the time period before 5000 years ago, i.e. preceding major demographic changes across sub-Saharan Africa. It also proposes to understand the LSA transition and a previously identified population cline that no longer exists between eastern and southern Africa. Current aDNA from sub-Saharan Africa is of a Holocene age. The results suggest that a long period of long distance exchange and population mixing in Africa was replaced by strongly structured local populations from around 20 ka onwards, until the spread of farming. These shed light on aspects of deep population history in Africa, but certain aspects of the paper either lack detail or are a little too technical for a broad reach paper.

I am not able to review the genetics components of this paper and leave this to a specialist in this subject area. However I evaluate the fit between the genetics and the archaeology and recommend some changes. With these changes, I recommend the paper for publication as an important contribution to the field.

We have edited the manuscript to clarify technical language and make it more broadly accessible, for example by ensuring that each paragraph of our results section communicates clearly the major finding described in that paragraph. In addition, we have significantly revised the introduction and discussion in order to reduce words, and in the process have made them clearer and more accessible. Due to space constraints, we are not able to elaborate further upon archaeological detail in the manuscript, but we have created a new Supplementary Note 2 that provides essential archaeological background.

Originality and Significance: The authors present new genome wide aDNA data with radiocarbon ages from three very Late Pleistocene and three early to mid-Holocene associated with LSA technology. Their

results extend the previously identified cline both spatially and temporally, and find further complexity within it. Critically, and this is where the novelty of the paper lies, they also show that there are other sources of for the observed complexity. They show that populations ancestral to the studied individuals had significant interaction and mixing across vast region spanning the eastern coast to the interior across eastern and south-central Africa. However, this section is somewhat convoluted and a little tricky to follow. The authors sometimes explain terms and hypotheses more clearly than at other times in the paper. It would be good if they went through this entire text and tried to make it a little clearer for general readership.

We have gone through the entire text carefully to make our findings clearer to a general audience, in particular shortening the introduction and discussion to focus on the most relevant points.

What is interesting about the results is that while many archaeologists assume long-distance contact between foraging groups, these results suggest that, at least for the time period studied which post-dates ca. 80-60 ka, this was not the case across as vast region. That said, they do hypothesise that prior to the emergence of the three-way cline, long-range dispersals and significant admixture did take place at some point in the Pleistocene, likely before 50 ka and certainly before 20 ka. I would like to see some elaboration on this in the paper. I understand it is a hypothesis that might be over-stretching the data a bit, but I think it is nevertheless important to set up more comprehensively. These populations became strongly structured again around 20 ka, remaining strongly local until the expansion of farming in Africa. I would like to see a little more elaboration here on key features of the LSA that support this argument. Currently it is a little light on detail and generic, given the paper starts off with saying it is shedding light on the LSA transition.

Our results show that long-range mobility and interaction took place starting 80 ka (with the development of the eastern to southern African genetic cline) and further after 50 ka (with the development of a three-way cline with central African ancestry). By 20 ka we see a decline in long-range interactions and an increase in 'living locally.' We have clarified this result in the manuscript. Our discussion includes consideration of this result in terms of LSA archaeological and recent historical linguistic data, and our new Supplementary Note 2 does so in more detail than would be possible in the main manuscript. In particular, we note that while the archaeological evidence for increased regionalization (i.e., living locally) in the LSA is not always clear and is sometimes conflicting, our new genetic results tip the balance in favor of those lines of evidence that suggest reduced scale of social networks that involved reproduction.

Data and Methods: The SI details on excavation and sampling are clear and well written. The radiocarbon ages are not all what could be described as precise and some individuals could be younger than stated, which is acknowledged by the authors. Other ages are indirect. For the oldest individual, where a date of 18.5ka is given in the main text, this is really a maximum age and I note there is a stratigraphic inversion at the site. This individual too, could be a little younger. Nevertheless, the authors present convincing arguments for the approximate ages given, and any deviations from the estimates are unlikely to exceed a couple of thousand years at most for the Pleistocene, and I doubt more than about one thousand for the Holocene. I would like to see the main paper make clear which age estimates were maximum estimates, and couple these with minimum estimates to give the reader a better understanding of these age brackets, particular when so many of the dates are problematic.

We agree that transparency on radiocarbon dating is important and we have provided detail in the supplement to that effect. In the main manuscript (lines 129-131), we provide the full ranges for the indirectly-dated individuals and refer the reader to Supplementary Note 3 and Supplementary Table S2 for further details.

Conclusions: I think the authors do a good job of not over-stretching their results and make clear what is a hypotheses requiring more work, and what is well supported by the data they offer. I cannot comment on the genetics-specific analytical work and its quality.

We have been careful not to overinterpret our findings in the discussion section, and we hope this paper will help lay the groundwork for future integrated genetic, archaeological, and linguistic studies.

References: No doubt a few further archaeological references will be added with the recommended revisions, however the references seem adequate.

We have reduced the number of references in the main text due to space constraints, but many of the cut references are now preserved in the supplement, with the creation of the new Supplementary Note 2, which provides archaeological background and interpretation.

Clarity/Context: In the section on uniparental markers, please explain Y chromosome haplogroups as you do for mtDNA haplogroups. What is the significance of chrY haplogroup B2? The text talking about ‘calling’ Y chromosome, etc. also reads oddly to the general reader. If this is a technical term, I recommend changing it to something more intuitive for Nature’s general readership.

In lines 151-152, we have clarified the language around Y chromosome haplogroups; in particular we clarify the significance of the B2 haplogroup (that it is widespread); and we explain that our data show several haplogroup lineages were more widespread in the past than they are today. We have maintained ‘calling’ as the standard term for this analysis.

More generally, it would be helpful if some technical terms such as ‘excess allele sharing’ were explained since the paper is for a broad readership across different disciplines. As mentioned above, it would be helpful if the methods and results sections were a little clearer and less technical. It would be really helpful if there was a brief summary in there about all the population groups the authors speak about and what is known about them. Currently information is a little distributed within the text. Please also check your estimates for the beginning of the LSA, and this is important. They do not match the evidence from Panga Ya Saidi.

We have clarified terms wherever possible in methods and results. For example, regarding excess allele-sharing, in lines 185-189 we clarify the meaning of this (degree of relatedness) and refer the reader to methods:

*We used allele-sharing tests (f -statistics; **Methods**) to investigate further which individuals had differences in their degree of relatedness to either ancient South African foragers (*AncSA*; **Table 1**), the Mota individual, or present-day Mbuti. Consistent with the PCA, most pairs of individuals from the same region (including from different time points) were nearly symmetric in their ancestry ($|Z| < 3$; **Supplementary Table S7**).*

We have also reviewed the rest of the manuscript text with an eye to making the paper accessible while still communicating essential information. We have moved some of the more technical details of methods and results to the supplement.

Regarding a brief summary of present-day groups (e.g., Ju’hoansi, Mbuti, Agaw) considered in this study, we refer the reader to Supplementary Note 1, which provides some information on terminology and choices of groups used in the analyses. Since we use only published DNA data for present-day groups, the most reliable and detailed information about these groups can be found in the original publications (and references cited therein).

Regarding the beginning of the LSA and the role of the site of Panga ya Saidi in Kenya:

The sequence for this site suggests a gradual transition with limited evidence of lithic production techniques traditionally associated with the LSA until relatively late. Miniaturization begins around 67 ka, but other LSA signatures such as backed artifacts appear only around 54-48 ka and disappear for millennia after this (Shipton et al 2021). The work of d’Errico et al. (2020) at PYS also demonstrates that the earliest potentially symbolic objects are ambiguous (e.g., ochre, pierced shells). D’Errico et al state that the earliest clear evidence for interest in ochre is at 48.5 ka, and most evidence of ornamentation postdates 33 ka.

We do not find that the publication of PYS changes the date presented in our manuscript. We are careful to treat the LSA as a confluence of new cultural traditions, not as a time period with a fixed starting and ending date. We cite d'Errico et al 2020 (ref. 4) and Shipton et al 2021 (ref. 5) in the paper, stating on lines 82-85:

By ~50, technological innovations and symbolic behaviours (e.g., ornaments, bone tools, pigments, microliths) present earlier in the Middle Stone Age (MSA) become more consistently expressed across sub-Saharan Africa^{3,4}. Archaeologists refer to this phenomenon as the transition to the Later Stone Age (LSA)^{1,4-6}. By ~20 ka, these material culture components are nearly ubiquitous.

Referee #2 (Remarks to the Author):

In order to investigate the population history of Late Pleistocene/Holocene African foragers, Lipson et al. generated and analyzed genome-wide data of six completely new ancient modern humans from sub-Saharan Africa and improved the quality of data of 16 previously published ancient individuals (most notably the Mota individual for who they now report genome wide data to 25x coverage). The data generated represent an exciting contribution which should be of interest for a wide range of researchers in the field of human population genetics and evolution. Moreover, by successfully recovering ancient DNA from 3 Late Pleistocene individuals they are doubling the time depth of ancient human DNA reported from sub-Saharan Africa. The study demonstrates well that although the conditions of ancient DNA preservation in Africa are not optimal (about 1/5th of the specimens screened show evidence of ancient DNA preservation, and median coverage is 0.06x), there are at least some sites and remains that can be made to work well enough.

We appreciate the reviewer's positive feedback and enthusiasm for our contribution and its interest to a broad range of researchers across disciplines.

However, the study doesn't really provide substantial new outcomes that would change the way we think about ancient foragers populations in Africa in the late Pleistocene/Holocene. Basically, it replicates and investigates further previous findings that showed a gradient of relatedness between eastern and southern African foragers during the Holocene, highlights the presence of central African-related ancestry in ancient eastern and south-central African foragers and provides evidence that this population structure existed already during the Late Pleistocene as early as 18 kya. The study shows that genetic similarity among ancient eastern African foragers correlates well with geographic proximity. The authors also calculate effective population sizes and show that these are in the range of what is known for African forager groups today.

We respectfully disagree that this study does not provide new outcomes. The novelty of this study – as noted by Reviewer 1 – is in demonstrating the complexity of a three-way cline that was not previously appreciated, and revealing for the first time key diachronic changes in this population structure and distinct regional trajectories, which we interpret with respect to archaeological evidence. The finding of central African ancestry throughout our sampled area – a substantial portion of eastern and south-central Africa – has not been previously reported and is important to understanding connections to an under-researched part of the continent.

General major comments

1. The inconsistency in the way the individuals are referred in the main text, figures, legends and supplementary materials (sometimes by geographic location, sometimes archaeological name, sometimes by Individual ID or by different combinations of these) makes the reading of the manuscript unpleasant and difficult to follow. In order to help readers interpret the results without having to keep looking up to relevant informations about the individuals in Table 1, I advise the authors to adopt a nomenclature similar to the one used in Skoglund et al. 2017 or to consistently give the estimated ages and geographic locations of individuals at relevant places in the text as for example for samples discussed between lines 177 and 189.

We have now gone through the manuscript, table and figures to make sure we are referring to individuals the same way throughout. We have opted to keep Individual ID as our primary designator, and not to tie a precise median date to each individual's ID tag, because several of the dates have wide ranges and/or are indirect or even – in some cases such as at Gishimangeda – entirely absent. Given these problems, such precision is in our opinion unwarranted and any dates incorporated into an individual ID may become “fossilized” into future publications about these individuals, even as new dates from the same site become available. However, we appreciate the suggestion of consistently giving estimated ages and locations of individuals throughout the text, and we have done this wherever appropriate, bearing in mind the constraints of word count in the main text.

2. Both mtDNA and nuclear sequences support the conclusion that individuals from each region sampled cluster with one another with few exceptions. However, the authors barely comment on specific individuals/populations context and whether there is an age-related signal here. Considering the age range of the samples (~20,345-17,025 cal BP to 500,320 cal BP), I advise the authors to integrate more systematically the time scale in their discussions.

Although we are limited by space constraints, we have added some consideration of whether individuals are buried together and at the same time or not on lines 294-295, and 370-372, with respect to the individuals from Malawi. We note that sample sizes are very small to support meaningful comparisons across time.

3. The range of values and consistency between methods of contamination estimates, as well as the potential effect of contamination on all results is not presented or discussed enough in the main text. While the authors reported that they estimated contamination using 3 different approaches, only the results of the heterozygosity rate at variable sites on the X chromosome (for males only) are presented Table S1. I would like to see the actual contamination estimates from each method for the samples. eg: What means “small amount of contamination” (Line 362) or “a non-negligible amount of contamination” (Line 924)? In the qpGraph analyses, two samples are modelled post-hoc as contaminated by present-day European DNA only in cases where a lack of fit lead the authors to attempt to include contamination in their estimates. What about potential effects of contamination on the PCA and *f*-statistics? I'm also worry about that possible contamination by present-day Africans may have gone unnoticed. The study requires additional estimates of contamination using existing approaches such as those based on linkage (Nakatsuka et al. 2020) or on damage (Meyer et al. 2016; Peyrégne and Peter 2020).

We agree that more detail was warranted for these points, and we have substantially extended our treatment of contamination estimation with a new Extended Data Figure 1, additions to **Supplementary Table S2**, and especially through discussion in the new **Supplementary Note 5**, all linked to a new sentence in the main text (lines 136-139). In addition to the X chromosome-based estimates, **Supplementary Table S2** provides damage rates, mtDNA match rates (similar to the X chromosome method), and now also directly computed sex chromosome ratio information. All of these results are discussed and contextualized in **Supplementary Note 5**. Based on our combination of approaches, we believe individuals I13763 and I2966 have detectable but modest (< 10%) contamination; among individuals with moderate to high coverage, I19529 and I10726 also have some limited evidence, but not as strongly as the former two, and with no indications that it affects our results.

In the presence of contamination, we would indeed expect effects on population genetic analyses such as PCA and *f*-statistics (the latter especially in the context of admixture graphs), whereby the affected individuals would artificially show genetic similarity to groups related to the contaminating individual(s). As we discuss in **Supplementary Notes 5-6**, allele-sharing with non-Africans in particular can be a powerful signal pointing to possible contamination, although we stress the importance of combining such signals with other lines of evidence to determine whether they reflect true ancestry or not. Contamination from present-day African individuals is possible, but given the high degree of divergence between the lineages of ancestry represented among the ancient foragers and among almost all people living today, we would expect any substantial contamination to create a noticeable signal in our results (PCA or other analyses), as well as through the metrics used to assess contamination (just as for any other source). Overall, the fact that we don't observe such signals or outliers (outside of the examples mentioned) serves as evidence for the limited-to-negligible effect of contamination on our results.

With the additions made in the revised version, we believe that our discussion on these points is now quite thorough. The main limitation of the methods used here is that they are less effective for the lower-coverage individuals, but the same is true as well for other methods, which also have their own additional limitations (e.g., requiring reference data for contamLD, and only being applicable to single-stranded libraries for authentiCT). Thus, we feel that the methods we have employed provide the best available characterization.

4. The statistics used in the study are state-of-the-art methods and are all appropriate. However, the description of uncertainties and probabilities values should be better reported in the figures or the figure legends. For example, for the PCA analysis it would be nice to report in the figure or the legend how much genetic variation each PC captures. Figure 1C doesn't have axes, which makes it very confusing at first. For the qpGraphs, the Z-score of the lowest/highest outlier should be reported in the figure or the legend.

For PCA, we have edited and hopefully clarified the figure, with only one plot displayed (see also our response to a subsequent comment below). Because of our “supervised” approach, we believe that the variance captured by the PCs is not a meaningful quantity for the individuals shown (projected) in the plot. For the admixture graphs, in addition to reporting worst outlier Z-scores in the main text, we also now give them in each of the respective supplementary figures.

Specific major comments

1. The analysis of ‘phenotypic SNPs’ presented line 193-195 should be a bit more detailed, as written it is not sure what message should the reader take home here? Are the data not good enough in any individual to know whether these alleles were present? Or were these mutations not yet present in any of these individuals. Given that there are some individuals with good coverage (e.g. Mota, who hasn't been included for these analyses according to Table S6) there must be some of the individuals where one can say for sure whether they carried these variants or not?

We have extended the analysis to more individuals and explained the results in more detail (for space reasons, the full discussion is provided in **Supplementary Note 5**). It is indeed true that some genotypes can be determined with confidence for high-coverage individuals (for example, the Mota individual has at least 17 reads for each site and no observed derived alleles).

2. Figure 2 is comprehensive only after reading the whole main text and supplementary Note 4. The legend should give more details to help the reader to understand it, especially the lower part. I suggest to the authors to clarify how they selected the different cluster, to include the proportion of the three-way admixture clade at the top of each cluster sub-graph and to indicate the lengths of the different branches in the sub-graphs. Indication of the age of the individuals should also help the readers to better interpret the results.

We have edited the figure and substantially revised the caption in the new version. Our goal with this figure was to strike a balance and provide a simple summary of the results without excessive detail (the complete graph with all admixture events, parameters, etc., is available separately for those readers who are interested) while also making sure the presentation is understandable (for which we appreciate the suggestions). We now show internal branch lengths and explain the meaning of the clusters. Our feeling is that including dates in the figure would not add much value, especially given the lack of clear temporal structure (and the fact that a number of individuals are undated); they are still available in **Table 1**. Mixture proportions can be found in **Extended Data Figure 4** and **Supplementary Table S9**, and we have also now moved the main-text figure showing proportions from each ancestry source so that it is adjacent to the admixture graph figure.

3. The qpGraph analyses supporting the inter and intra-regional relationship presented seem robust and were done intelligently, with a lot of discernment and care. The graph of Model 3 Figure S4 is particularly impressive and certainly required a lot of work. However, they require also a lot of time and focus to decipher, and most admixture events are impossible to trace visually in Model 2 and 3 (Figure S3, S4 and

S6). I think it will be easier to the readers to follow if the admixture analyses are presented in a more gradually manner. Following the strategy used to make the graph Figure S6, I suggest to the author to make independent graphs for each of the different cluster they identified (Kenya, Tanzania, Coastal/Island and Malawi/Zambia), to discuss them separately before putting the complete picture of the Model 3 graph together. This should help to better visualize and comprehend the contrast in structure between the Malawi/Zambia cluster and the Tanzania/Kenya cluster.

Also, the authors should clarify the choice of individuals used in the different graphs, for example why individuals from Malawi I4468 and I2967 are not included in the graph of the shared three-way admixture clade of the Malawi individuals presented figure S6? Using different colors for the different cluster of individuals should make the graph easier to read.

We acknowledge that the full admixture graph results can sometimes be difficult to digest for larger models, which is why we wished to present a simpler version in **Figure 2** (see previous comment). Our feeling remains that the easiest way to understand the inferred structure of admixture graph results is through that figure. We appreciate the other suggestions, and to aid in the interpretation of the full graph figures, we have now added separate color schemes for the different regions/clades and ancestry sources to help separate them visually.

In terms of the analyses themselves, we believe that one of the key aspects of the admixture graph methodology (at least as applied in our paper) is the ability to fit multiple populations of interest simultaneously and evaluate them in the model together. Thus, we feel that separate fitting by region would effectively only remove constraints from the model and thus would not add new information. The model shown in S6 is a different case, where we wished to experiment with an alternative structure for the individuals from Malawi (while keeping all other ancient individuals present as well), given the lack of shared drift signals between the different sites. The two other individuals mentioned are not included because S6 is designed as an alternative version of Model 2, so it should have the same set of individuals in order to make the results comparable. Both are lower-coverage individuals from the same sites and similar times as others in the model already (I4427 and I2966, respectively), which is why we left them out from Model 2 initially; in fact, they are inferred to form clades with those other individuals, as shown for example in **Figure 2**, so their presence or absence should not change the structure of the S6 model.

General minor comments

1. Although the number of genomes that are now available from late Pleistocene and Holocene African individuals is growing, the number of individuals is still small and the sampling density in time and space uneven and low. Tanzania and Kenya are referred to as East Africa and Malaysia and East Zambia as South-Central Africa. The authors should probably be a little more cautious about what insights into 'continental scale' population structure can be detected.

Clearly aDNA is in its early years in Africa and we will know much more in another decade. Nevertheless, this analysis truly does have a continental scale in that the individuals analyzed span nearly 1/3 of sub-Saharan Africa. The paper also does provide meaningful insights at regional level despite its limited sample size by examining important differences in interaction networks, even within the relatively small territory of Malawi, and we fully acknowledge that future work has the potential to reveal much more.

2. The authors successfully modelled the ancestries in ancient Eastern African foragers as deriving from three divergent source population and that there may possibly be an additional source contributing to the Mota individual from Ethiopia. It might be good for the authors to discuss the constraints here ; in regards of the limited sampling and reference set as well as the data quality, inferences can only be made about *these* foragers. The conclusion seems unlikely that there is only these three ancestry sources for all these populations (as illustrated by the Mota result). The idea that there may exist of as-yet unsampled populations that contribute to later Southern and Northeastern foragers is very interesting.

This is a good point; we have emphasized it on lines 205-206.

Specific minor comments

1. The introduction feels a bit flat and do not emphasized the exciting specific questions and previous findings related to prehistoric African population history, to the exception of the genetic cline between eastern and southern African foragers. The first paragraph lacks references and can be skipped since the most relevant information in it is repeated in the third paragraph. The second paragraph introduces in broad details the transition to the Later Stone age (LSA), for which I hardly see the relevance in regard to the main results of the study. I think it can be shortened.

We appreciate this critical read of the introduction and we have significantly cut and reshaped it, both to reduce words and improve clarity and emphasize what is exciting and new. We note that Reviewer 1 asked for more, not less, detail about the Later Stone Age and relevant archaeology. Much of the cut detail has been moved to our new Supplementary Note 2, which provides more extensive archaeological background.

2. The author inverted and rotated the PCA plot presented in figure 1C, I assume in order to match the geographic distribution presented in Figure 1A. I found it very confusing, and I haven't understood at first what was the motivations. I suggest figure 1B and 1C to be presented in the same orientation, whether it is inverted and rotated or not.

We appreciate and agree with this comment and have now revised the figure to have only two panels: the map, and the full PCA in the orientation that matches the map. We have also edited the legend of the PCA for clarity.

3. The present-day individuals/populations used to compute the principal component of the PCA analysis are not reported in the plot, I am curious to see how they positioned compared to the ancient individuals.

Please find attached on the last page of this document a version of the PCA plot with additional populations added (bottom three in the legend). In order to enable a proper comparison with the other individuals shown, we used projected data for different individuals from the same groups used to compute the axes, with the exception that instead of Dinka (for whom we did not have data for additional individuals at hand), we used Mursi, a closely related Nilotic-speaking group. All three fall at the expected extremes of the plot.

4. The unit for related ancestry found in ancient African foragers in figure 5 is not reported.

This has been added to the caption.

5. The conclusion is a bit bland. I think the authors should rewrite it, they should remove the summary of the discussion and emphasize better the context and the relevance of their results. They should give more specific perspectives about the future of ancient DNA for the study of prehistoric Africa.

The discussion and conclusion have been significantly revised and the concluding paragraph almost completely rewritten. In it, we emphasize the importance of ancient DNA research in Africa and the future potential of this field.

6. Supplementary Note 1 and 3 are inverted.

This has been corrected.

7. The archaeological site summaries part in Supplementary Note 2 is very inconsistent between sites. For Fingira and Hora 1 sites the description focus more on the archaeological context and the dating of the samples while for Kalembe, Kisese II and Mlambalasi rockshelters we have in additional information about the site. I suggest changing the title "Archaeological site summaries" of that section of Supplementary Note 2 to "Samples archaeological context and dating", as I'm afraid part of the audience would skip that section thinking it is only about the sites.

We thank the reviewer for noticing this and we have renamed this part of the supplement (now Supplementary Note 3) as: "Details of sampled archaeological skeletons". A reason for providing different

kinds of archaeological context and information is that in some cases this data is published and can be found elsewhere, and in others it cannot; in general, we provide more detail where such information would be difficult or impossible for the reader to find elsewhere.

8. The authors should clarify the relevance of the intermediate admixture graph Model 2.

The three main purposes of Model 2 are (1) for its own sake, to provide an intermediate admixture graph version with a size and data quality level in between those of the relatively small Model 1 and quite large Model 3; (2) to enable follow-up analyses with such an intermediate-scale model; and (3) to aid in construction of the full Model 3, whereby we can repeat the add-one analysis using Model 2 as a base to help find signals of excess shared drift. An overview of our fitting procedure is now included in the **Methods**, and we have added clarification on these points in the relevant part of **Supplementary Note 6**.

9. In Supplementary Note 4, in order to explain their expanded models, the authors described intermediate qpgraphs which are not presented anywhere (e.g. lines 679 to 689, 691 to 700 or 825 to 835). It is very difficult to apprehend the graph only from writing description. I suggest to the authors to present every intermediate graphs described.

We appreciate the suggestion, but we feel that most of the additional admixture graph-based results are (a) highly specific and technical, and (b) only constitute minor variations from the main models. Thus, we believe that showing the full graphs for all of them would be of limited value. We did add a new supplementary figure for the model including Hadza and Sandawe to ensure that all of the primary model extensions from the main text of the paper are represented by visualizations.

10. In Supplementary Note 4, I think the section “Relationships of the three primary forager ancestry sources to other sampled populations” should be presented before the section “Detailed modeling of forager individuals”. It is more intuitive to read first about the big picture before going to details.

This change has been made.

Reviewer Reports on the First Revision:

Referee #2 (Remarks to the Author):

I reviewed the initial submission of this manuscript, and the edited version is greatly improved in clarity.

The authors addressed my concerns by modifying and restructuring the main text to provide clarifications or by adding Supplementary notes. When the concerns couldn't be fully addressed the authors clearly explained the limitations associated with the issues and provided logical arguments to justify their choices.

That authors' interpretations are logical, and their conclusions remain solid. I do not detect any scientific flaws in the manuscript which I think is acceptable for publication.

Referee #3 (Remarks to the Author):

I was invited as a reviewer to take a particularly close look at the archaeology side of the manuscript (and the archaeology/genetics intersection) and the changes that were made in regard to the previous comments by original reviewer #1 (I was not part of the original referees).

Having gone in detail through the manuscript, supplementary information and reply to reviewers, I can attest that the authors have conformed to all of reviewer #1's original criticism and constructive suggestions. Some of this can be found in the main text – with many clarifications and a style that is now good to follow – with many details on the archaeology 'hidden' in the supplements (I understand that the authors need to do so due to space constraints, yet it remains a shame that archaeology gets so little room in such papers; this is not a critique aimed at the authors).

Particularly, I like the careful take of the authors on the 'origins' of the LSA which really is only there at all/most sites by 20 ka (before the evidence is sporadic, heterogenous and often up to debate) plus the thoughtful use of language as explained in "Supplementary Note 1. Terminology chosen in this study".

From my own, first-time reading of this manuscript, I applaud the authors for an important and timely paper which presents important new information and yet does not overreach when it comes to implications. The key results were clearly summarized, the findings are novel and highly significant, and references, tables and figures are appropriate. I would hope for an even more inter-disciplinary approach of combining archaeological and genetic evidence in the future for an even more holistic narrative (particularly giving the vast archaeological information more space), but this goes beyond what the current paper can achieve. In my opinion the paper will make a good fit with Nature and attract much interest from within and outside the archaeology and genetics scientific community.

Author Rebuttals to First Revision:

N/A